

# 4D Tracer Flow Reconstruction in Fractured Rock through Borehole GPR Monitoring

Peter-Lasse Giertzuch[1], Joseph Doetsch[1,2], Alexis Shakas[1], Mohammadreza Jalali[3], Bernard Brixel[1], and Hansruedi Maurer[1]

[1]Institute of Geophysics, ETH Zurich, Zurich, Switzerland
[2]Lufthansa Industry Solutions, Raunheim, Germany
[3]Chair of Engineering Geology and Hydrogeology, RWTH Aachen, Aachen, Germany

**Correspondence:** Peter-Lasse Giertzuch (peter-lasse.giertzuch@erdw.ethz.ch)

**Abstract.** Two borehole ground penetrating radar (GPR) surveys were conducted during saline tracer injection experiments in fully-saturated crystalline rock at the Grimsel Test Site in Switzerland. The saline tracer is characterized by an increased electrical conductivity in comparison to formation water. It was injected under steady state flow conditions into the rock mass that features sub-mm fracture apertures. The GPR surveys were designed as time-lapse reflection GPR from separate

boreholes and a time-lapse transmission survey between the two boreholes. The local increase in conductivity, introduced by the injected tracer, was captured by GPR in terms of reflectivity increase for the reflection surveys, and attenuation increase for the transmission survey. Data processing and difference imaging was used to extract the tracer signal in the reflection surveys, despite the presence of multiple static reflectors that could shadow the tracer reflection. The transmission survey was analyzed by a difference attenuation inversion scheme, targeting conductivity changes in the tomography plane. By combining the time-

lapse difference reflection images, it was possible to reconstruct and visualize the tracer propagation in 3D. This was achieved by calculating the potential radially-symmetric tracer reflection locations in each survey and determining their intersections, to delineate the possible tracer locations. Localization ambiguity imposed by the lack of a third borehole for a full triangulation was reduced by including the attenuation tomography results into the analysis. The resulting tracer flow reconstruction was found to be in good agreement with data from conductivity sensors in multiple observation locations in the experiment volume

and gave a realistic visualization of the hydrological processes during the tracer experiments. Our methodology proved to be successful for characterizing flow paths related with geothermal reservoirs in crystalline rocks, but it can be transferred in a straightforward manner to other applications, such as radioactive repository monitoring or civil engineering projects.

## 1 Introduction

Flow and transport processes in fractured rock have been a key focus of basic hydrogeological research and are relevant for nu-

merous applications and research fields. These include risk assessment of contaminants (e.g., Andričević and Cvetković, 1996), nuclear waste disposal (Cvetkovic et al., 2004) and the exploitation of deep geothermal energy (DGE) (Brown et al., 2012). Virtually all fluid transport in granitic crystalline rock is carried by discrete permeable fractures that are connected within a fracture network. Field investigations in such complex subsurface environments are extremely challenging, as no complete





direct observations of the fracture geometries and hydrological processes can be made. Conventional hydraulic and tracer tests
only provide spatially discrete observations, such that flow and transport properties have to be interpolated or upscaled between
observation points, or they have to be estimated with numerical simulations and simplifying assumptions. Therefore, the com-
plex models needed for flow and transport in a discrete fracture network (DFN) can often only be weakly constrained by data.
Additionally, the choice of conceptual model introduces other simplifications, such as parallel plate approximations and radial
flow assumptions (Council, 1996). Therefore, generalized assumptions, based on local observations within the subsurface, lead
to significant uncertainties. For example, current models may encounter difficulties in estimating the effective surface area
for heat exchange in natural fracture networks, which is highly relevant for DGE applications (de La Bernardie et al., 2019).
Evidently, a better understanding of flow and transport in fracture networks outside of the lab-scale is necessary (Amann et al.,
2018), but the possibilities to locate and visualize flow paths within a fractured crystalline rock volume are limited.

Ground penetrating radar (GPR) and GPR-responsive tracers are a possible option to monitor and visualize such processes.
GPR makes use of electromagnetic waves in MHz to GHz frequency ranges. As for any EM wave, the propagation in the
subsurface is primarily dependent on the dielectric permittivity $\epsilon$ and the electrical conductivity $\sigma$ of the host medium. While
$\epsilon$ primarily affects the propagation velocity, $\sigma$ controls primarily the wave attenuation. Depending on these subsurface param-
eters and the frequencies employed, GPR can be used for penetration depths extending from the centimeter range to hundreds
of meters. The initial GPR pulse is reflected and/or transmitted at interfaces of contrasting electromagnetic parameters. The
resulting signal can be recorded and estimations about the subsurface properties can be made (e.g. Jol, 2009).

GPR-responsive tracers that create a local contrast can be detected by comparing repeated GPR measurements during tracer
and in-situ water flow experiments. Such time-lapse GPR surveys are not as established as, for example, time-lapse electrical
resistivity tomography (ERT) or time-lapse seismic imaging, but they can be useful in engineering and hydrology, as the elec-
trical properties, sensed by GPR, can be affected by state variables of interest. Brewster and Annan (1994) were able to monitor
dense nonaqueous phase liquids (DNAPL) spills, by exploiting their low relative dielectric permittivity (compared with the pore
water). Also, the propagation of water in unsaturated soil was successfully monitored with GPR (e.g. Allroggen et al., 2017).
Salt water within a saturated environment with low formation water conductivity introduces a local signal attenuation that was
successfully detected in several cross-hole transmission GPR studies (Niva et al., 1988; Day-Lewis et al., 2003). Besides con-
ducting transmission measurements, it is also possible to consider reflected GPR waves for detecting conductivity-changing
tracers. The reflection from a thin conductive layer can be similar to a permittivity boundary, with a reflection coefficient and
a phase shift dependent on the conductivity (Lázaro-Mancilla and Gómez-Treviño, 1996; Tsoflias and Becker, 2008).

If the tracer-induced change of the electromagnetic properties is small, as expected for small-aperture fracture networks or
low tracer to formation water contrast, more elaborate difference approaches are needed to extract information. This has been
demonstrated successfully for tracer tests in granite with reflection GPR (Dorn et al., 2011; Shakas et al., 2016). In theory, the
difference between the reference and the monitoring data should contain only the tracer-induced changes, and static features
from unchanged reflections should cancel out. However, for identifying subtle changes in the volume of interest, it is required
to ensure a very high consistency and repeatability of the different data sets. Recently, an effective data processing procedure
has been developed by Giertzuch et al. (2020) that allowed for saline tracer signal extraction in a host rock with several reflec-





tors and fracture apertures in the sub-mm range.

Unfortunately, there is no straightforward link between tracer-induced signal changes in GPR data and the governing material properties. Problems are caused by the azimuthal symmetry of GPR antennas, which makes it difficult to unambiguously identify the location of a reflector. Likewise, in the presence of a heterogeneous (with regards to GPR wave propagation) host rock, it can be difficult to relate travel times and amplitudes, obtained from transmission experiments, with the spatial distributions of the electrical permittivity and conductivity.

Some of these problems can be alleviated when the results of reflection and transmission surveys are combined. In this contribution, we present results from a study, in which such a combined analysis allowed the 3D tracer flow path to be reconstructed. The approach is based on differencing schemes for both, the reflection and transmission data. After providing a description of the experimental setup, we present our processing workflows for the difference reflection imaging and the difference attenuation tomography. Finally, we outline how the resulting information can be combined to reconstruct the 3D tracer flow path
through the experiment volume.

## 2  Test Site and Experimental Setup

### 2.1  Grimsel Test Site

The Grimsel Test Site (GTS) is an underground rock laboratory in Switzerland. It is located in a weakly fractured rock mass in the central Alps (Figure 1). It is operated by the National Cooperative for the Disposal of Radioactive Waste (NAGRA) and
has been used for a range of research projects[1]. The experiments presented here were conducted in the scope of the In-situ Stimulation and Circulation (ISC) experiment (Doetsch et al., 2018; Amann et al., 2018; Krietsch et al., 2018). They were performed in the southern part of the GTS, located at a depth of about 480 m below the surface. The hydraulic heads indicate that the experiment volume and its fracture network are fluid-saturated. Figure 2 shows a geological model of the relevant part of the GTS, based on the work of Krietsch et al. (2018). The main features that influence fluid flow in the experimental
volume include two brittle-ductile shear zones (S3, shown green in Figure 2). They intersect the AU tunnel (shown blue in Figure 2) between four geophysical monitoring boreholes (GEO1 to GEO4), two of which (GEO1 and GEO3) were used for the GPR surveys. In total, 15 boreholes were drilled in the project volume to characterize the subsurface conditions. Besides the geophysical monitoring and the injection boreholes, these include three strain monitoring, three pressure monitoring, and three stress characterization boreholes (Doetsch et al., 2018). From this set of boreholes, only the ones relevant for this experiment are
depicted in Figure 2, namely the injection borehole (INJ2, green), the geophysical monitoring boreholes (GEO1 and GEO3, blue), and three pressure monitoring boreholes (PRP1-3, magenta). The PRP borehole intervals indicated in black contain permeable fractures and were permanently isolated. The tracer injection interval (INJ2.4), indicated as a red sphere at a borehole depth of 23.38 m, was isolated with straddle packers.

---

[1] www.grimsel.com



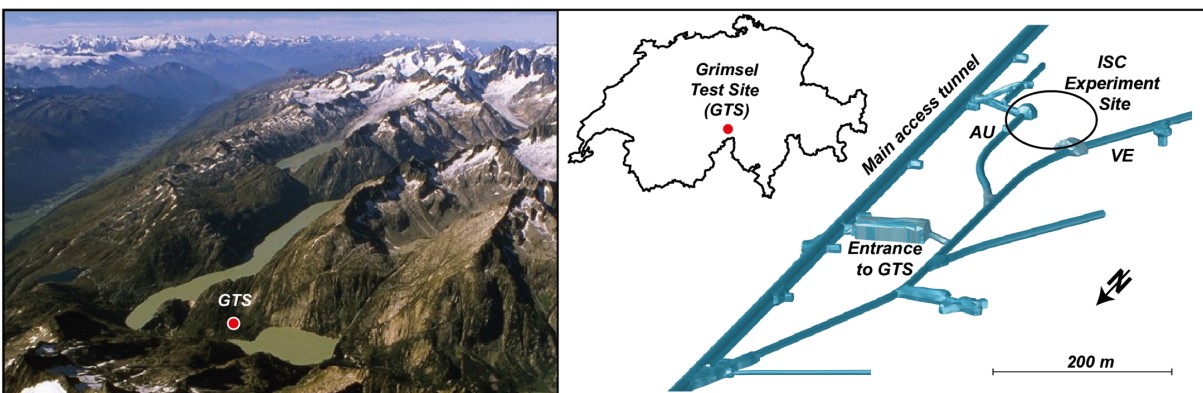

**Figure 1.** GTS located in the central Alps in Switzerland. The ISC experiment was conducted in the southern part of the GTS, between the VE and the AU tunnels. Figure from Doetsch et al. (2017).

## 2.2 Grimsel ISC experiment

The controlled ISC experiment was carried out to investigate the seismo-hydromechanical processes during the creation of a geothermal reservoir. It included two stimulation phases, featuring hydraulic shearing and hydraulic fracturing that were conducted in February and May 2017, respectively (Amann et al., 2018). A comprehensive characterization for the relevant part of the GTS was crucial for a detailed understanding of the stimulation effects in the subsurface and a subsequent transfer of knowledge into reservoir creation. Therefore, additionally to the monitoring performed during the stimulations, pre- and

post-stimulation characterizations were carried out with various geological, hydrogeological, and geophysical methods. A special emphasis was put on the fracture geometries and permeabilities, with the aim of a detailed flow field and fracture network reconstruction. Aside from GPR, as described here, by Giertzuch et al. (2020) and by Doetsch et al. (2020), extensive hydrologic testing was performed, tracer tests were run, and borehole logs were acquired, e.g. (Brixel et al., 2017; Jalali et al., 2018a; Krietsch et al., 2018; Kittilä et al., 2019; Brixel et al., 2020a,b; Kittilä et al., 2020a,b).

The fracture density in the host rock varies between 0 and 3 fractures per meter and increases to >20 fractures per meter between the S3 shear zones (Krietsch et al., 2018). Doetsch et al. (2020) found a decrease of seismic velocity between the S3 shear zones and were able to link this to a known increase of fracture density and permeability in this region (Krietsch et al., 2018; Brixel et al., 2020a). In the same study, surface GPR was used to identify the S1 and S3 shear zones, but experienced low contrast for the S3 shear zones, due to the perpendicular orientation towards the survey tunnel. The GPR propagation velocity

was found to be approximately $0.12 \, \mathrm{m/ns}$.

Fluid flow was shown to be primarily fault-controlled in well testing experiments by Brixel et al. (2020b), who found permeability to be strongly decreasing (from $10^{-13} \, \mathrm{m^2}$ to $10^{-21} \, \mathrm{m^2}$) within $1 \, \mathrm{m}$ to $5 \, \mathrm{m}$ from the fault cores. The S3 shear zones appear to have feature extension fractures that are linking the two S3 shear zones and create highly permeable cross-fault connections as shown in Brixel et al. (2020b). Additionally to the fault-controlled nature of the flow system, a drainage effect

of the AU tunnel plays a key role for fluid flow at the ISC test site (Jalali et al., 2018b; Kittilä et al., 2020b). From dye and

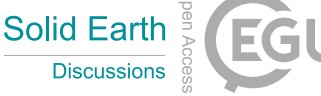

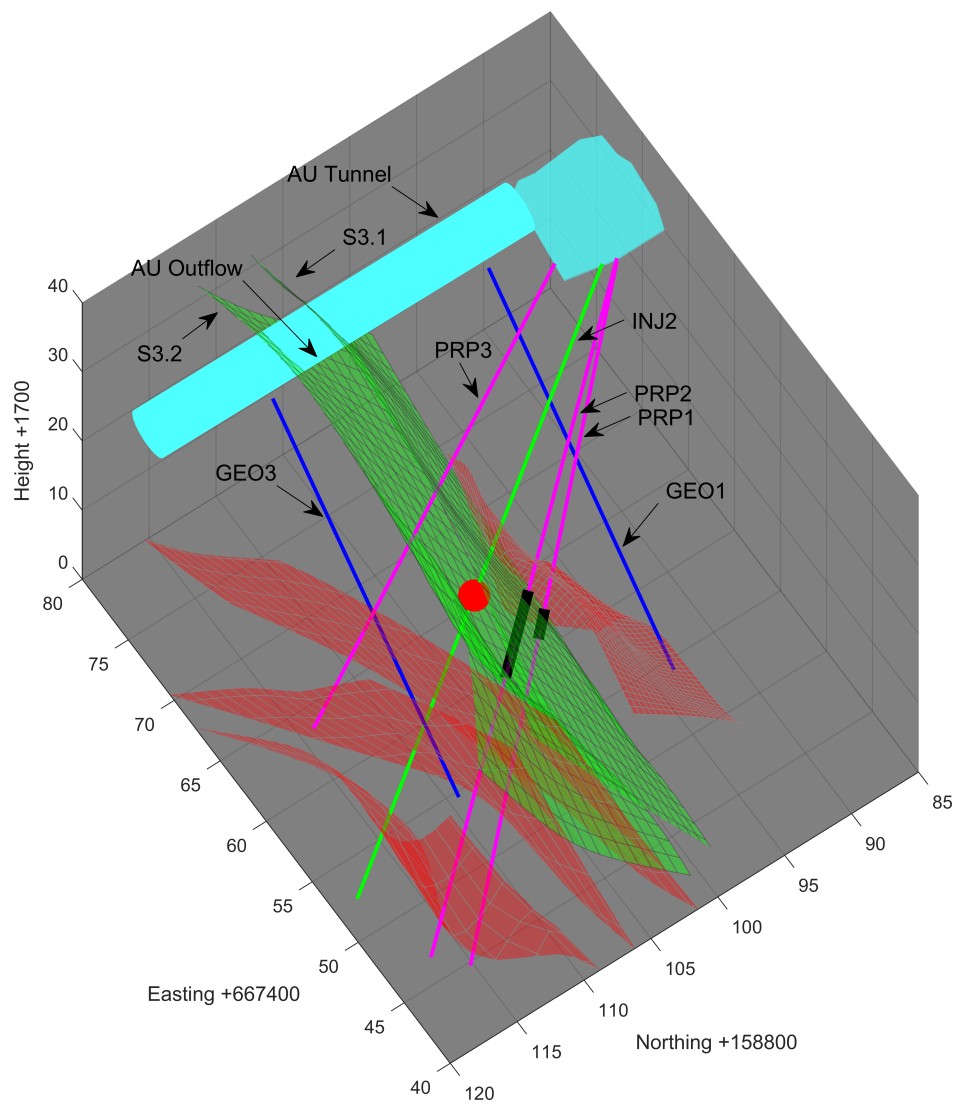

**Figure 2.** Geological model of the experiment volume with relevant structures and boreholes. The S1 and S3 shear zones are shown in red and green, respectively. The GPR survey was performed in the blue GEO boreholes. The salt tracer injection point is indicated as a red sphere. Monitoring intervals in the PRP boreholes are indicated in black.

saline tracer tests in the INJ2.4 interval connections towards PRP1.3, PRP2.2 and the AU tunnel are known with relatively fast breakthroughs in that order (Kittilä et al., 2020b,a; Giertzuch et al., 2020; Jalali et al., 2019).

In a first-order estimate on hydraulic packer tests of single fractures, Brixel et al. (2020a) calculate equivalent hydraulic apertures to range from 2 to 130 µm, with a mean of 30 µm. However, this calculation relies on a parallel plate assumption with
smooth fracture walls, hence locally the true fracture apertures may diverge significantly from this estimate.





## 2.3 GPR Tracer Experiments

Two GPR tracer experiments were conducted in November and December 2017. Each experiment relied on the same tracer and injection location, and they had a similar monitoring setup. A saline tracer with a conductivity of approximately 60 mS/cm was injected at a constant flow rate of approximately 2 L/min in the INJ2.4 interval, which is located in between the S3 shear zones (Figure 2). The formation water showed a conductivity of around 80 μS/cm. Conductivity data loggers were connected to the AU outflow, and to the intervals PRP1.3 and PRP2.2. Before and after tracer injection, formation water was injected continuously for several days at the same location and flow rate to ensure a steady flow state. For the first experiment, 100 L of saline tracer were injected over the course of 50 minutes, while for the second experiment the volume was doubled to 200 L, and the injection was performed over 100 minutes. The second experiment was performed during an ongoing heat injection experiment. While the hydraulic properties of the individual flow paths appeared to be affected by this (Kittilä et al., 2020a), the general flow path geometry can be assumed to have remained unchanged.

In general, the experiments were designed to be very similar to the experiment described and evaluated by Giertzuch et al. (2020). The main differences included a larger injection interval and therefore higher injection rates, a higher tracer conductivity, the use of a salt-water tracer instead of a salt-water-ethanol tracer, a different GPR acquisition console, and slightly different GPR settings.

## 2.4 GPR Data Acquisition

In total, we acquired three GPR data sets, two of them during the tracer experiments and one transmission GPR survey in the unperturbed experiment volume. In all of the described GPR experiments, MALÅ[2] 250 MHz GPR borehole antennas were used. The MALÅ CU2 control unit was equipped with a multichannel module to allow the connection of up to 4 antennas.

During tracer experiment 1 (100 L injection), GPR data were acquired as a reflection survey with transmitter and receiver antennas placed in the GEO3 borehole (Figure 3b). The antenna separation was held constant at 1.76 m, and every 10 cm a trace was recorded. The distance was measured with a trigger wheel, and the measurements were recorded while pulling the antenna array upwards, to ensure constant cable tension. The GPR survey lasted for 5 hours, but it was partly interrupted due to empty antenna batteries. In total, 38 usable reflection profiles were recorded.

Tracer experiment 2 (200 L injection) was carried out as a combined reflection and transmission survey with four 250 MHz antennas connected as seen in Figure 3c. One *moving* antenna set of transmitter and receiver was operated in borehole GEO1 with a fixed distance of 1.76 m. Another *static* set was used in GEO3 with a fixed distance of 5 m. The trigger wheel was used at GEO1. This setup allowed for a reflection survey carried out in GEO1, during which (as in the previous test) every 10 cm a trace was recorded, again triggered during an upwards motion of the antenna array.

Simultaneously, the two transmission channels (see Figure 3c) were triggered every 20 cm, with the *static* antenna array in GEO3 fixed in position. After each of these recordings the antenna array in GEO3 was moved to a new position, to generate another full transmission data set. In total, eight different *static* antenna array positions were occupied in GEO3. The positions

---

[2]Guideline Geo (MALÅ Geosciences), Malå, Sweden





of the single antennas in the *static* array were chosen to be placed alternating in two sets, as presented in Table 1. The straight

ray patterns of the two deployments are shown in Figures 4b and 4c. With this procedure it was possible, to use the data either

as a more comprehensive data set with all 16 positions but a lower time resolution, or with only 8 positions but a higher time

resolution. Both of these options were later used to invert for the change in attenuation.

|  | Set 1 positions [m] | | | | Set 2 positions [m] | | | |
|---|---|---|---|---|---|---|---|---|
| Antenna 1 | 0 | 2.5 | 10 | 12.5 | 5 | 7.5 | 15 | 17.5 |
| Antenna 2 | 5 | 7.5 | 15 | 17.5 | 10 | 12.5 | 20 | 22.5 |

**Table 1.** Deployment scheme of the static antenna array in GEO3 for the transmission survey. The positions are referenced in m from the bottom of the borehole.

The acquisition of each of the (full) transmission data sets took about 40 minutes and during that time, 8 reflection profiles

were recorded simultaneously. The GPR survey lasted for eight hours, but during that time the antennas had to be recharged.

In total, one of the two transmission channels recorded eight full sets, the other seven. Additionally, one full transmission data

set was recorded prior to the tracer injection to serve as the *reference* data (Figure 3c).

For a more detailed analysis of the experiment volume, an additional, more comprehensive cross-hole data set, subsequently

referred as baseline data set, was acquired with the same antennas when the experiment volume was unperturbed by any tracer.

This data set was acquired with a single antenna set and the MALÅ ProEx console. The static antenna was positioned at every

meter in borehole GEO3, and the moving antenna was moved along borehole GEO1 with a trigger interval of 20 cm. The setup

can be seen in Figure 3a and the associated straight ray patterns are shown in Figure 4a.

## 3   Methods

The processing of both, the reflection and transmission GPR data sets, can be subdivided into two parts. By means of a

baseline processing of the reference data sets, acquired prior to the injections, static images of the test volume prior to the

tracer injections can be derived. With the subsequently applied difference processing, temporal changes between the individual

measurements can be analyzed.

### 3.1   Baseline Reflection Imaging

The processing of the reference reflection data sets included relatively few steps. It started with a DC-shift removal, followed

by frequency filtering using a 10-50-350-450 MHz Kaiser bandpass window. Then, a singular value decomposition (SVD)

filter (e.g., Press, 2002) was applied to remove the direct wave and spurious system ringing effects (removal of eigenvectors

associated with the largest eigenvalue). To enhance the amplitudes of later phases, a time-varying gain was applied, consisting

of a linear gain proportional to time $t$ to account for spherical spreading and an exponential gain to account for attenuation.

Finally, the reflection sections underwent a Kirchhoff migration using a constant velocity of 0.12 m/ns that was confirmed by

the tomography results, other GPR surveys at the test site (Giertzuch et al., 2020; Doetsch et al., 2020).





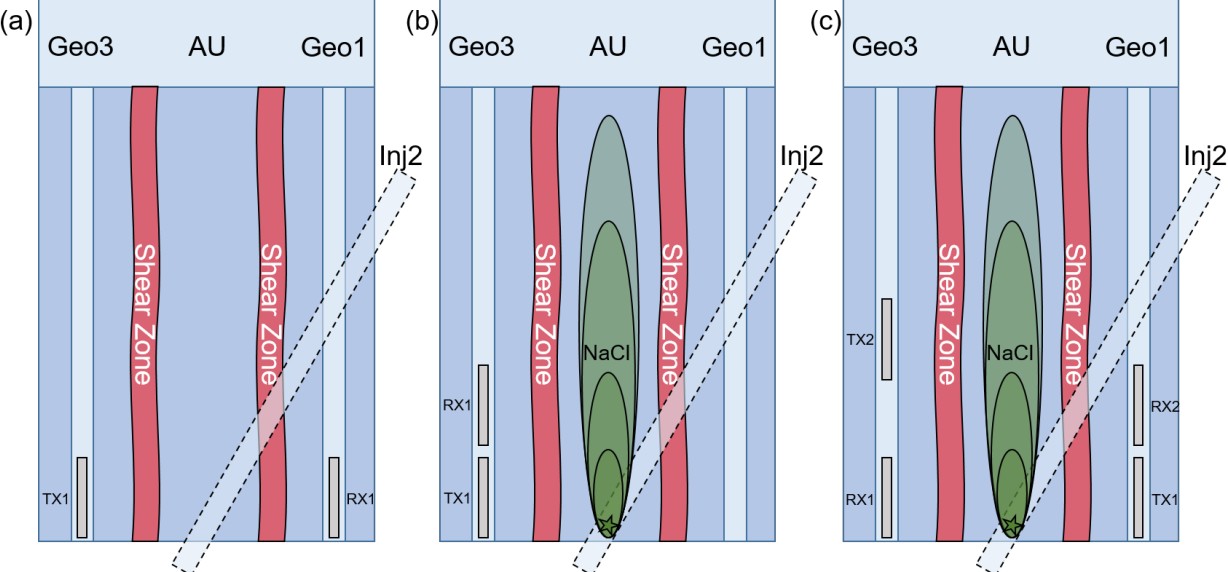

**Figure 3.** Schematic of the acquisition setups for the GPR surveys performed. a) Setup for baseline data set without tracer injection. b) Setup for tracer experiment 1 (reflection survey in borehole GEO3). c) Setup for tracer experiment 2 (combined transmission and reflection survey).

## 3.2 Difference Reflection Imaging

The tracer presence in the subsurface is expected to introduce only a minor change in the GPR signals. This is primarily due to the small fracture apertures found in the investigated volume. Therefore, a differencing approach was required to illuminate the signature of the tracer. In principle, calculating the difference between the monitoring and the reference data sets should leave only changes introduced by the tracer, whereas static reflections cancel out. The success of such a procedure is highly dependent on the consistency between the reference and monitoring data sets. Especially in survey areas with several reflectors,

as expected in the experiment described here, minor repeatability errors will lead to improper cancellation of the various reflections and may obscure the tracer signal.

Our difference reflection processing follows closely the approach described in detail by Giertzuch et al. (2020). As for the baseline processing, a DC shift removal was initially applied to all data sets. Then, a temporal trace alignment was performed to ensure a proper consistency between monitoring and reference data sets. The alignment was performed by calculating a cross-

correlation between the reference and monitoring traces and shifting them according to the largest correlation. By up-sampling the traces by a factor of ten, a sub sampling interval alignment was possible. After temporal alignment, some elements of the baseline processing were applied to the reference and monitoring data sets (frequency filtering, SVD filtering and application of a time-varying gain).

Next, a spatial alignment procedure was applied. During acquisition, antenna position inaccuracies may have occurred. Such

errors may arise from cable slip, twist, and stretch, trigger wheel inaccuracies, or handling mistakes. We applied a cross-





correlation and trace interpolation based testing scheme for identifying and correcting positioning errors. In the GEO3 data set only minor corrections were required, but in the GEO1 data set, shifts up to 10cm had to be applied. The origin of these inconsistencies is unknown.

After applying the processing steps described above, some traces showed artifacts in form of spiky signals that originated likely from electronic cross-talk with other instruments on site. They were removed by replacing them with linearly interpolated values in time and space from adjacent samples and traces. Then, the reference data set was subtracted from the monitoring data sets, which resulted in the GPR difference profiles.

Despite the extensive correction procedures, the difference profiles still exhibited minor artifacts, resulting from improper canceling of static reflections and diffractions. They were suppressed by using again a SVD filter, with which the eigenvectors associated with the largest five eigenvalues were removed. We found this approach to be more effective than eliminating further singular values in the undifferenced data.

As for the baseline reflection processing, a time-domain Kirchhoff-migration was then applied to the difference section. Finally, a temporal smoothing filter was used to suppress minor random variations between the profiles.

Contrary to what Friedt (2017) and Giertzuch et al. (2020) have described, we did not encounter significant sampling rate variations or drifts. The greater stability in sampling rate can be likely attributed to the use of the older MALÅ console *CU2* instead of the MALÅ *ProEx* model.

### 3.3 Baseline Tomography

For carrying out the travel time inversions of the baseline transmission data, several pre-processing steps were performed.

– *Determination of the T0-time (emission of the transmitter pulse)*
   The T0 time was estimated by analyzing in-air measurements taken at antenna separations between 3 m and 16 m.

– *Sampling rate drift correction*
   The *ProEx* console, employed for the transmission measurements, is known to be prone to sampling rate drifts. We accounted for this by acquiring a zero-offset-profile (ZOP) prior to the survey. Assuming that the sampling rate remained stable during the relatively short acquisition time of the ZOP, the tomographic (multi-offset) data could be corrected by comparing the travel times of the ZOP with the corresponding traces in the multi-offset data.

– *Anisotropy correction*
   The recorded cross-hole data showed a slight anisotropy. Since the anisotropy is not expected to vary significantly within the tomographic plane, we judged it appropriate to apply an anisotropy correction procedure, as described in Maurer et al. (2006), and to subsequently employ an isotropic inversion code.

For carrying out the inversions, we considered a ray-based inversion algorithm as described by Maurer et al. (1998), and for the subsequent amplitude inversions, we were following the approach described by Holliger et al. (2001). To account for the under-determined component of the inversion problem, suitable damping and smoothing constraints were applied.





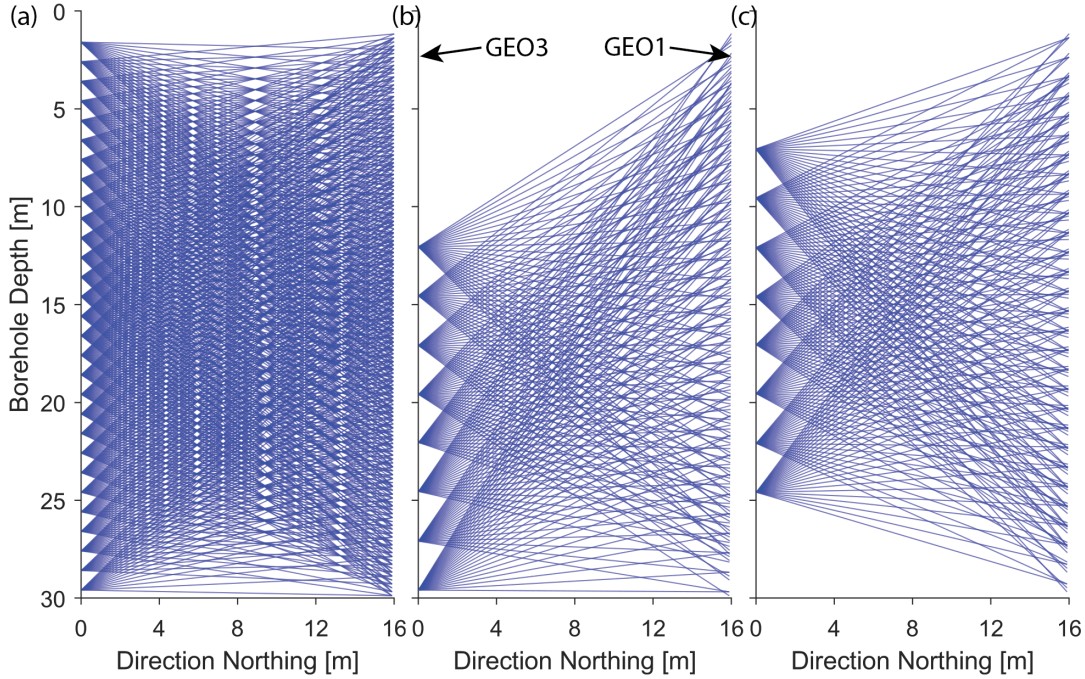

**Figure 4.** Ray coverage for the transmission data sets. Only every fifth ray is plotted. a) shows the comprehensive baseline data set, b) and c) show the two subsets for the time-lapse transmission experiment.

### 3.4 Difference Attenuation Tomography

The presence of a conductive fluid, such as a saline tracer, leads primarily to GPR signal amplitude attenuation, and only

225 marginal changes of the travel times of GPR waves. Therefore, we focus here on attenuation tomography. The relative changes of the GPR amplitudes between the reference and monitoring transmission data sets are expected to be relatively small. Therefore, we developed a difference tomography approach, with which minor amplitude variations can be exploited. To asses the amplitude differences, introduced by the presence of the tracer, the monitoring and reference data sets needed to be pre-processed in a consistent manner, such that a valid comparison was possible. For that purpose, we applied the same DC-shift

230 correction as for the reflection data and aligned the monitoring to the reference data with the same temporal trace alignment as described for the reflection data. Additionally, we removed outliers, resulting from faulty traces either in the reference or monitoring data sets.

In a next step, we calculated *amplitude difference factors* $a$ for each source-receiver pair. They were obtained by performing a linear fit of the first 30 sampling points after the onset of the first arriving wave train:

235 $$E_n^m = a \cdot E_n^r \quad \text{with} \quad n = 1,..,30 \quad , \tag{1}$$





where $E^r$ is a reference trace and $E^m$ is the corresponding monitoring trace. 30 sampling points correspond approximately to the first wave cycle. Therefore, out-of-plane effects, such as reflections, were largely excluded from our analysis.

From the baseline tomography, the ray path length distribution $L_{ijk}$ ($i = 1...$no. of sources, $j = 1...$no. of receivers, $k = 1...$no. of inversion cells) was available. As outlined in Holliger et al. (2001), ray-based amplitudes can be computed using :

$$E_{ij} = \frac{A_0 \Gamma_i \Theta_j e^{-\sum_k \alpha_k L_{ijk}}}{\sum_k L_{ijk}}, \tag{2}$$

where $A_0$ is the source strength, $\Gamma_i$ is the transmitter radiation pattern (and coupling), $\Theta_j$ is receiver radiation pattern (and coupling), $\alpha_k$ is the attenuation in the $k$th inversion cell. Accordingly, the difference factor $a_{ij}$ can be written as

$$a_{ij} = \frac{E_{ij}^m}{E_{ij}^r} = \frac{A_0 \Gamma_i \Theta_j e^{-\sum_k \alpha_k^m L_{ijk}}}{A_0 \Gamma_i \Theta_j e^{-\sum_k \alpha_k^r L_{ijk}}} = \frac{e^{-\sum_k \alpha_k^m L_{ijk}}}{e^{-\sum_k \alpha_k^r L_{ijk}}}, \tag{3}$$

where we distinguish now between the attenuations in the reference and monitoring data sets ($\alpha_k^r$ and $\alpha_k^m$) and assuming that the ray path distribution $L_{ijk}$ does not change significantly between experiments. As can be seen in Equation 3, the unknown source strength, the radiation patterns, and the antenna coupling cancel out. Applying the logarithm to Equation 3 results in

$$-\log(a_{ij}) = -\log\left(\frac{e^{-\sum_k \alpha_k^m L_{ijk}}}{e^{-\sum_k \alpha_k^r L_{ijk}}}\right) = \sum_k \alpha_k^m L_{ijk} - \sum_k \alpha_k^r L_{ijk}. \tag{4}$$

Setting $\hat{a} = \hat{a}_{ij} = -\log(a_{ij})$ and $\boldsymbol{\Delta\alpha} = \Delta\alpha_k = \alpha_k^m - \alpha_k^r$ allows Equation 4 to be rewritten in a compact matrix vector form as

$$\hat{a} = \mathbf{L}\boldsymbol{\Delta\alpha}. \tag{5}$$

This represents a linear system of equations that can be solved with a suitable algorithm. As for the baseline tomography, this system of equations includes an underdetermined component, which requires regularization in terms of damping and smoothing.

We applied a two-step inversion scheme on our data. As stated before, the transmission data recording scheme allowed for either the use of a full data set with all 16 positions, but a lower time resolution, or a higher time resolution with only 8 positions. First, we considered the larger 16 position data sets to generate the difference attenuation results, whereby the regularization constraints minimized the magnitudes of $\boldsymbol{\Delta\alpha}$. Then, we inverted the smaller 8 position data sets that featured a higher temporal resolution, whereby the regularization constraints penalized deviations of $\boldsymbol{\Delta\alpha}$. This procedure gave results at the high temporal resolution, practically with the spatial resolution of the larger data set. As longer ray paths over the diagonal source-receiver combinations tend to be less reliable due to radiation characteristics and signal attenuation, we introduced a weighting factor on $a$ in the inversion that was inversely proportional to the ray path lengths, serving as an error estimate.





## 4 Results

### 4.1 Baseline Reflection Imaging

Figure 5a presents the processed reference profiles for the GEO3 survey and Figure 5b shows the corresponding profile for the GEO1 survey. Both profiles show numerous overlapping reflections. To aid the visualization of natural (faults) and artificial (boreholes) features, also for the difference images, the most important reflections for these surveys are highlighted with dashed lines. They include reflections from the S3 shear zones, the Injection (INJ2) borehole, the PRP1 and PRP2 boreholes, as well as the other visible GEO boreholes. To ensure a proper identification of the different reflections, the expected reflection GPR responses for the different boreholes were computed using the migration velocity of $0.12\,\mathrm{m/ns}$ and compared to the measured reflections. Additionally, the interpretation was based on previous GPR surveys from these boreholes with higher spatial resolution, as shown in (Giertzuch et al., 2020). Further strong reflections that are not highlighted in the figures could also be identified, but were left out here for the sake of image clarity. Both reference profiles show that the relevant regions in the subsurface can be imaged by the GPR surveys. The injection interval in INJ2 that is located between the S3 shear zones, and also the PRP boreholes that are known to show breakthroughs for this tracer experiment are visible.

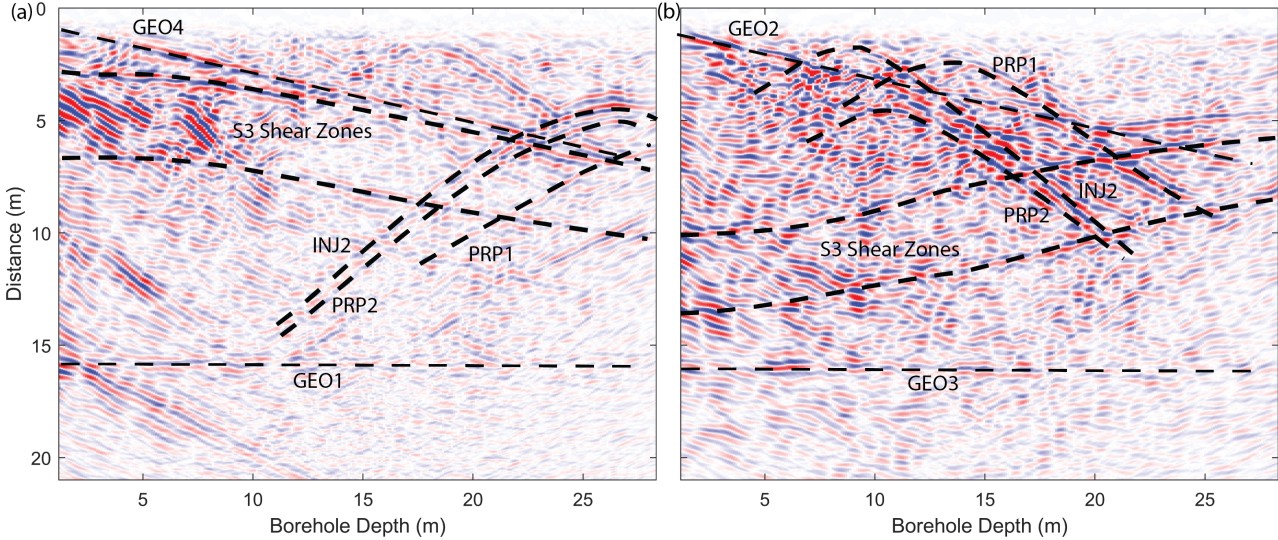

**Figure 5.** Reference profiles for the reflection GPR surveys. a) Reference from GEO3. b) Reference from GEO1. Important reflections are indicated with dashed lines.

### 4.2 Difference Reflection Imaging

The tracer injection started for both experiments at $t = 0\,\mathrm{min}$ and lasted until $t = 50\,\mathrm{min}$ for the GEO3 survey and $t = 100\,\mathrm{min}$ for the GEO1 survey. In Figure 6, four difference reflection profiles for both of the experiments are presented that were recorded at similar times. The visible tracer reflections are delimited by solid green lines. Additionally, calculated reflection response





areas are indicated in the graphs that correspond to relevant borehole intervals and the AU outflow.

In Figure 6a)-d), the difference reflection profiles are presented for the first experiment with reflection GPR monitoring from
the GEO3 borehole. At $t = 12$ min after the start of the tracer injection, a clear tracer signal around the injection point is visible.
Parts of the tracer already appear to have moved towards the PRP1.3 interval.

At $t = 41$ min, the tracer signal is expressed more strongly, as more tracer has been injected. It also started to propagate away
from the injection point towards the AU tunnel, as the reflections in the image show. Apparently, the tracer has split up at the
injection location to propagate towards PRP1.3 and towards the AU tunnel.

At $t = 126$ min, the tracer response around the injection point has disappeared, as formation water was injected after the tracer.
The tracer's signature is now visible close to the AU tunnel, but around a borehole depth of 13 m, the tracer signal strongly
reduced or lacking. At $t = 264$ min, the tracer reflection appears strongly around the AU outflow point in the AU tunnel.

Figure 6e)-h) shows four difference reflection profiles for the second experiment with reflection GPR monitoring from the
GEO1 borehole. At $t = 12$ min, a clear tracer signal around the injection point is again visible. Parts of the tracer already appear
to have moved towards the PRP1.3 interval.

At $t = 45$ min, the tracer signal appears again stronger, as more tracer has been injected. Again, it appears to have split in two
directions and also started to propagate away from the injection point towards the AU tunnel.

At $t = 125$ min, the tracer has traveled further towards the AU tunnel. Two flow paths seem to be visible around the injection
point and merge at approximately 15 m borehole depth and near the PRP1.3 interval. The reflection signal around a borehole
depth of 13 m is only barely visible. At $t = 264$ min, the tracer appears to have reached the AU outflow point. The area around
the injection point has now mostly cleared up. Again, formation water was injected after the tracer injection.

In both surveys also the calculated reflection positions from the PRP2.2 interval are reached. This is in good accordance with
the conductivity measurements at this interval that showed a breakthrough here at approximately 100 min after tracer injection.
However, the radial distance of PRP2.2 to the survey boreholes is similar to the radial distance of the injection point to the
survey borehole. Therefore, the reflections around PRP2.2 and INJ2 can not be well distinguished, which introduces some
uncertainty here.

Time-lapse videos of these reflection survey results are available in the supplementary material.





**Figure 6.** Time-Lapse difference reflection imaging results. a)-d) show the results for the experiment conducted in the GEO3 borehole at four different time steps. e)-h) show the results at corresponding times for the experiment conducted in the GEO1 borehole. The identified tracer reflections are highlighted in green, while the regions with lower or lacking tracer reflection is marked in red.





## 4.3 Baseline Velocity and Attenuation Tomography

Figure 7 shows the inversion results for the velocity and the attenuation of the GPR signal in the tomography plane. Additionally, the S1 and S3 shear zones are depicted within the plane for orientation. No dominant structures are visible in the velocity and attenuation images. The velocities vary slightly around approximately 0.12 m/ns, thereby justifying the migration velocity used during the reflection processing. The attenuation is generally low, which can be expected for a granitic host rock.

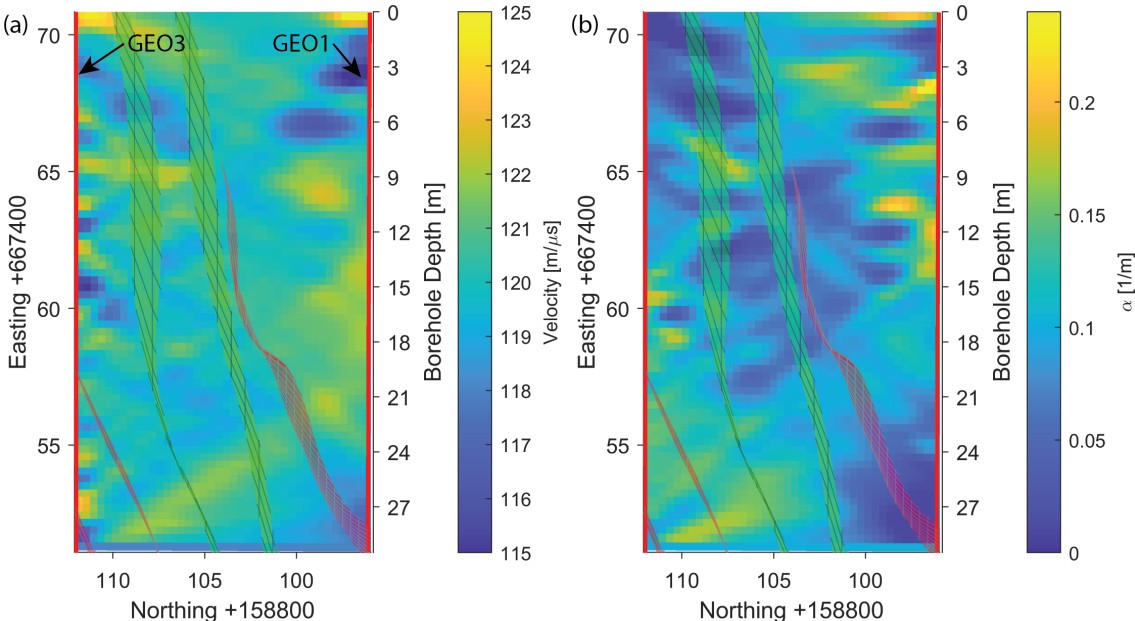

**Figure 7.** Results of the baseline inversion. a) shows the velocity result, while b) shows the attenuation. GEO1 and GEO3 are shown in red, the S1 (red) and S3 (green) shear zones are also depicted for orientation.


## 4.4 Difference Attenuation Tomography

The difference attenuation tomography results are presented in Figure 8 at time steps corresponding to those shown for the difference images in Figure 6. Already 20 min after tracer injection, there is a clear increase in attenuation at a depth of approximately 21 m, labelled with A. This corresponds to an area close to the injection location, which is slightly below the
inversion plane. Therefore, the tracer likely traveled upwards through the inversion plane.

At $t = 50$ min, the tracer signal is more pronounced and extends around the injection location, but no tracer propagation can be seen. After $t = 125$ min, another region with increased attenuation appears at a depth of approximately 8 m, labelled with B, and close to the GEO3 borehole at a distance of around 4 m. Later during the experiment, ($t = 270$ min), the attenuation around the injection point (A) appears reduced, but the attenuation around location B has further increased. This corresponds to an expected tracer appearance close to the AU tunnel outflow. Unfortunately, no antenna positions above 7.5 m were available





(Figure 4). Therefore the attenuation increase could not be traced further upwards.

The inversion could be fitted to a root-mean-square (RMS) value of approximately 0.05, while the amplitude difference factor can range in theory between 0 and 1. In our data, the factor mainly ranged between 0.7 and 1, however some data points showed an increased factor of >1. This would correspond to a decrease in attenuation, which was not allowed in the inversion routine,

therefore these data were not fitted higher than 1, and are thus increasing the RMS value slightly. The inversions were run for ten iterations. A time-lapse video of the difference attenuation tomography results is available in the supplementary material.

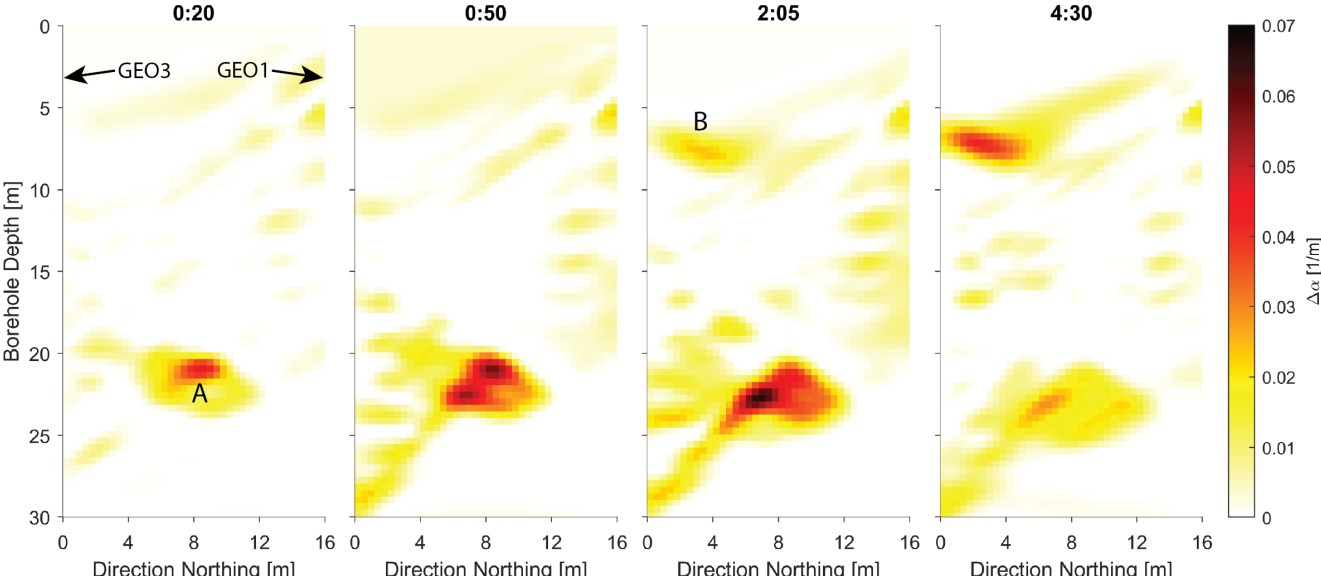

**Figure 8.** Results of the difference attenuation tomography for different time steps after tracer injection. The tracer-induced rise in attenuation can be seen early around a borehole depth of 22 m (labelled A) and later at around 8 m (labelled B).

## 5    3D Flow Path Reconstruction

The results shown in Figure 6, suffer from an azimuthal ambiguity. Meaning, the distances away from the boreholes including the GPR antennas are known, but the azimuthal angle from the borehole axis is unknown. For every time step, as shown in

Figure 6, we could therefore calculate a hollow *tubular* shape around the borehole axis on which the reflection must have originated. The width (along the distance in Figure 6) of the reflection patterns translates into the *wall thickness* of the tube, while its length and location along the borehole is defined by the patterns length and location (along the borehole depth in Figure 6). It is noteworthy, that the extent of these tubes does not represent the actual extent of the reflector, but only the extent of possible reflector locations.






The tracer flow path geometry can be assumed to be identical in both experiments, such that a reflection, visible in both experiments, should have occurred at the same locations in the experiment volume. Therefore, we combined the results from the two reflection surveys to at least partially overcome the radial ambiguity and confine the tracer localization: The *reflection tubes* for both experiments could be calculated, and the reflection must have occurred at the intersection of both cylinders.

In theory, more than one intersection of two cylinders is possible, and this ambiguity cannot be resolved by using only two boreholes. However, as shown in Figure 9a, in this study the tubes were mostly just intersecting each other once. Only in the region that also coincides with the reduced tracer reflection in Figure 6 and the lack of attenuation between feature A and B in Figure 8, the tubes overlap further, resulting in two intersections, one above the GEO plane and one below.

While the tracer flow path geometry can be assumed to be identical for both experiments, the temporal behavior of the tracer propagation differs due to the different injection protocols. For example, in the first experiment, formation water replaced the tracer already after 50 min and reduces the tracer reflection around the injection point (Figure 6c). At this time, the reflection is still visible in the second experiment (see Figure 6g), as the injection lasted for 100 min. To address this, we first combined the reflections for all time steps to generate static tubes of the full tracer flow path for both experiments and analyzed their

intersection, as shown in Figure 9b. The lacking reflection response in the middle of the tracer flow in the GEO3 reflection experiment was estimated and manually interpolated. These estimations were based additionally on the results from Giertzuch et al. (2020), as the experiments were very similar and no major changes in the flow path were expected. The double intersection, described above, can be seen, as the reconstruction shows the tracer above and/or below the GEO3 plane in the middle of the path that extends towards the AU tunnel.


Once the flow path location is known, the temporal information for both experiments can be analyzed by intersecting the tube sections of a single time step from one experiment with the whole flow path information from the other experiment. This allows generating two time-lapse 3D reconstructions of the tracer propagation for the two experiments. This temporal evolution is presented for the second experiment in Figure 10a-d along with the results from the difference attenuation tomography. The

full time-lapse of the reconstruction for both experiments can be found in the supplementary material.

As can be seen in Figure 9a, the intersection of tubular shapes results in a significant location uncertainty. To provide additional constraints, the attenuation tomography results were included in the analysis. The tomographic images are superimposed in Figure 10. The tracer is expected to manifest itself in form of increased attenuation. In the reflection data, the tracer signa-

ture splits around the injection point into two branches of propagation, one in the direction of the AU tunnel and one in the direction of PRP1.3 (Figure 6a,b,e,f). Therefore, it can be concluded that the branch propagating towards PRP1.3 is associated with the attenuation increase at feature A in Figure 8. The branch propagating towards the AU tunnel is assumed to stay below the tomographic plane until breaching through it, thereby causing an attenuation increase at feature B in Figure 8 and further propagating to the AU tunnel outflow.

The resulting tracer flow path is visualized in Figure 11. Figure 11a shows again the flow path geometry (blue) and the flow





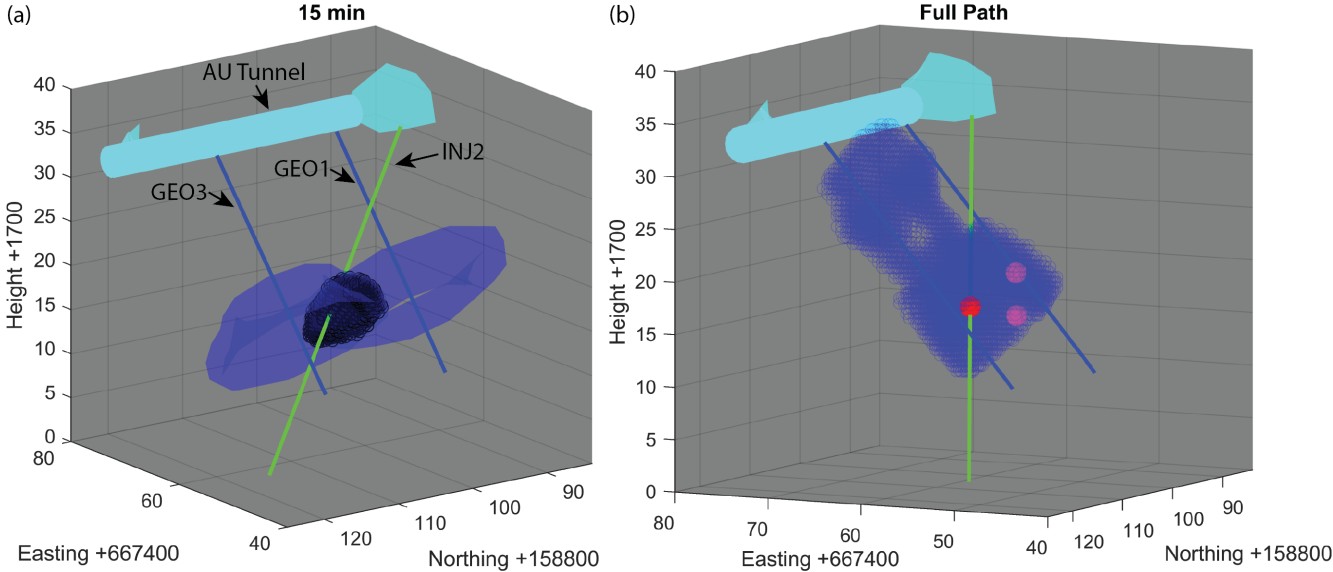

**Figure 9.** Tracer flow path reconstruction and important boreholes (GEO in blue, INJ2 in green). a) Reflection tubes at 15 min after tracer injection for both surveys in blue and their intersection in black. b) Fully reconstructed flow path for all time steps. The injection interval is indicated with a red sphere, the two PRP breakthrough locations are indicated as purple spheres.

directions (black arrows). Additionally, areas, where tracer flow seems to be blocked, are indicated by X. The labelled locations A and B refer to the areas of attenuation increase in Figure 8, indicating a passing through the GEO plane. The relation of the flow path reconstruction with geological features is visualized in Figure 11b. It can be inferred that the tracer traveled from the injection location first towards the PRP1.3 interval and the northern S3.2 shear zone, and from there propagated further along

this shear zone towards the tunnel outflow, as indicated with the black arrows.

## 6 Discussion

### 6.1 Interpretation of our Results

As shown in Figure 11, the GPR measurements allow the general tracer flow path regions and flow directions to be delineated. The 3D reconstruction based on the reflection tubes, however, does not represent the tracer position and its extent, but should

be understood rather as a volume, in which tracer reflections are likely to have occurred. Therefore not every position in the reconstruction was necessarily tracer filled at some point, but the tracer will have mostly flown within the extent of the reconstruction. By considering additional data and supplementary information, we will now make an attempt to further constrain the preferential flow path(s) and to validate our results. As shown in the difference reflection images in Figure 6, and also shown schematically in Figure 11, the tracer splits up at the injection point into two branches, one traveling towards PRP1.3 and

PRP2.2, and the other towards shear zone S3.2 and finally towards the AU tunnel. This could be also confirmed by conductiv-


(a) 00:21:00

(b) 00:48:00

(c) 02:05:00

(d) 04:28:00

**Figure 10.** Time-lapse tracer flow path reconstruction with tomography results at different time steps after tracer injection. The reconstructed tracer position is shown in blue, the tomography data is presented with the same color scale as in Figure 8 with a scaled transparency. The injection point lies slightly below the tomography plane.

ity meter data acquired in the PRP intervals (marked in Figures 2, 6 and 10). The breakthroughs of solute tracers in the PRP intervals and the AU tunnel is also described in dye tracer experiments (Kittilä et al., 2020b,a), to which our findings agree well. Interestingly, the attenuation tomograms, shown in Figures 8 and 10, do not show a continuous zone associated with the tracer flow, but they exhibit only patches of increased attenuation. This suggests that the tracer must have flown partially outside of
the tomographic plane from feature A to feature B (Figure 8). This is also confirmed by the 3D reconstruction visible in Figure 9b and 10. So, the question arises, if the flow path deviates below or above the tomographic plane. As the tracer appears to



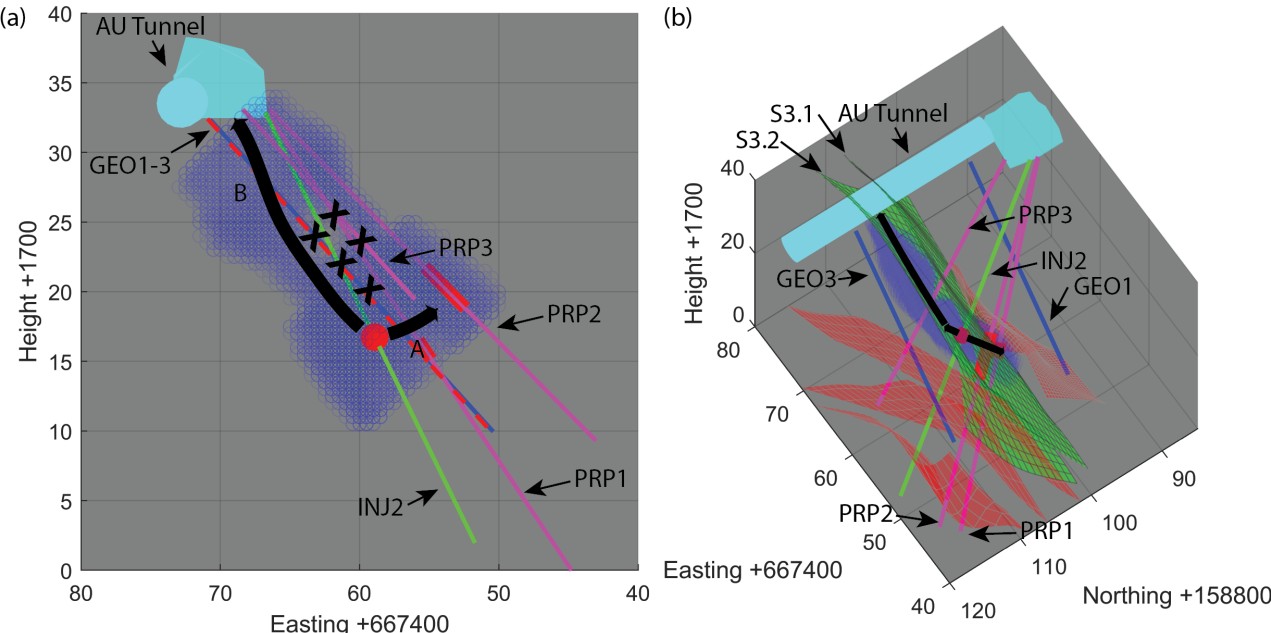

**Figure 11.** Interpretation of the tracer flow path reconstruction. a) Side view. The tracer indicated with the black arrows passes through the GEO plane (red/blue) at two locations, labelled A and B. b) Reconstructed flow path along with the shear zones S3 (green) and S1 (red) in the experiment volume. The PRP intervals are indicated in red.

split at the injection point, only the branch directed towards PRP1.3 is assumed to propagate through the plane, while the other branch apparently stays below the plane until reaching location B in Figure 11a. Borehole PRP3, which lies entirely above this plane and goes through S3.1 and S3.2 (see Figure 11), showed no outflow during the experiment, despite multiple identified

fractures in the borehole log, and can therefore bee seen as not connected during the timeframe and hydraulic configuration of this experiment. Therefore, we conclude that the tracer flow paths must lie below the tomographic plane. Figure 6 reveals a region of decreased tracer reflectivity in this flow path region (marked red). It is unclear, what caused such a decrease. We speculate that an unfavourable fracture orientation or strongly reduced fracture aperture could be the reason. Doetsch et al. (2020) also reported on an area between the S3 shear zones of strongly reduced reflectivity in surface GPR measurements.

While it is not entirely clear what causes this and the exact match of the locations is uncertain, these findings appear to be consistent.

Our tracer 3D flow path reconstruction agrees with the findings of Brixel et al. (2020a), who described the flow to be strongly fault-controlled, with single cross-fault connections between the S3 shear zones. Our results, however, indicate that the tracer

flow towards the tunnel happens along S3.2 and no flow is seen along S3.1. This observation has so far not been reported in previous publications within the scope of the ISC experiment. Therefore, the GPR experiments appear to not only serve as a visualization but also supply complementary refining information on the hydraulic system and provides an important additional





constrain on the flow and permeability architecture of the fracture system studied.

The GPR velocities $v$ and attenuations $\alpha$, obtained from the tomographic inversions, can be converted to dielectric permitivity $\epsilon$ and electrical conductivities $\sigma$ using the equations:

$$\epsilon = \frac{1}{v^2 \mu} - \frac{\alpha^2}{\omega^2 \mu}, \tag{6}$$

$$\sigma = \omega \epsilon \sqrt{\left( \frac{2\alpha^2}{\omega^2 \mu \epsilon} + 1 \right)^2 - 1}, \tag{7}$$

where $\omega$ denotes the angular frequency of the GPR signals, and $\mu$ is the magnetic permeability. This allows the attenuation changes, shown in Figure 8, to be converted to variations of the electrical conductivity. However, the tomography resolution and the necessary regularization makes it impossible to visualize small fractures in the results, as the variations in conductivity will appear smeared out over a larger area. Assuming that the blurred area of attenuation increase in the tomography near the injection point (feature A) is circular with a radius of 2.2 m (estimated from Figure 8), the average attenuation (from Figure

7) in this area would correspond to a conductivity of about 1.22 mS/m. The maximum mean attenuation in this area from the difference attenuation inversion was found at around 105 min, thereby matching well with the injection protocol. The conductivity for this increase was calculated to be 1.73 mS/m, hence an increase of 0.51 mS/m. The tracer itself has a conductivity of 6000 mS/m. Therefore, the maximum conductivity difference corresponds to only 0.0085 % of the tracer conductivity. To simplify, we assume that the tracer has traveled through a single fracture and the inversion plane cut through this fracture as a

cross-section. Therefore, we can calculate what fraction of the circular feature would need to be tracer filled with 6000 mS/m to appear as an average conductivity increase of 0.51 mS/m. This fraction of the circle area, that was found to be 0.0013 m², can be seen as the cross-section of the fracture. Taking the length of this cross-section as the diameter, we can calculate the apparent fracture aperture to be 0.3 mm. This concept is visualized in Figure 12. From the conductivity meter in PRP1.3, we know that the maximum conductivity measured there was not at 6000 mS/m, but only about 3000 mS/m. Using this value for

the calculation would result in an aperture of 0.6 mm. We certainly do not claim that this back-of-the-envelope calculation is a quantitative approach for determining the fracture aperture, but the apertures obtained are realistic. This is an indication that our attenuation tomograms are also realistic.

## 6.2   Broader Implications of our Research

The methods described here should be transferable to a variety of research fields and experiment sites, where detailed process localization is necessary and direct observations are not an option. A successful application primarily depends on the host medium, the accessibility and scales of the research site: the electrical conductivity of the host rock needs to be sufficiently low for signal propagation, and for using saline tracers a sufficiently strong contrast towards formation water needs to be ensured.




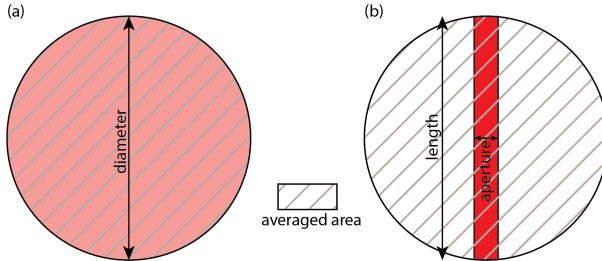

**Figure 12.** Concept of the fracture aperture estimation from the averaged attenuation circle. a) Averaged conductivity circle. b) calculated fracture estimate. For both the average over the circle is equal.

Moreover, different GPR monitoring locations are needed for the localization. Two boreholes where used in our study, but a
combination of surface GPR in access tunnels, or a combination of surface and borehole GPR could be possible. Furthermore,
the research site and the processes monitored must be resolvable by GPR, hence the scale of both, the site and the processes,
need to be considered.

Our methodology was developed within the scope of geothermal energy research, on an intermediate scale (deca-meter) be-
tween laboratory and full-scale application. While a transfer towards full-scale deep geothermal energy exploitation will be
challenging, application within hecto-meter scale research projects are reasonable. Such a project is for example currently
on-going in the Bedretto Laboratory in Switzerland (Gischig et al., 2020). Such intermediate-scale projects are employed to
investigate the complex seismo-hydromechanical interaction and therefore depend on a detailed flow system characterization.
Further, Shakas et al. (2020) have recently used 100 MHz borehole antennas for time-lapse difference GPR to monitor aperture
changes (and thus permeability changes) during hydraulic stimulation, thereby proving that the requirements for difference
imaging can be met on such scales. Our spatial reconstruction approach could be transferred to such experiments, where in-
stead of tracer fluid propagation the geomechanical changes could be monitored and localized by time-lapse GPR.

Of special interest could be experiments, where substantial changes of the flow paths can be expected, such as in pre- and post-
stimulation comparisons for reservoir creation. The methods described could provide a valuable tool to assess and visualize
how and to what extent the subsurface fluid flow was affected by stimulation.

We judge our methodology to be applicable for further applications, where flow paths or their changes are of relevance. Exam-
ples include tracing of fluid contaminants, related with radioactive waste repositories (such as the Äspö Hard Rock Laboratory
(e.g. Cosma et al., 2001)), or civil engineering projects. Our tracer reconstruction also shows that fluid flow is confined to a
limited number of fractures (compared to the total fractures encountered during drilling). Such information can therefore help
parameterizing site-specific numerical models of flow and transport in crystalline rock, where identifying permeable fractures
that control the flow behavior of the rock is important.
# 7   Conclusions

In this contribution, we have presented results from two borehole GPR experiments that were performed to monitor saline tracer flow in weakly fractured crystalline rock with small apertures. Both experiments had a similar tracer injection protocol, but they were used to acquire complementary data sets. A geometrical reconstruction approach was applied that successfully

overcame the radial ambiguity in the data and resulted in a 3D tracer flow path reconstruction. This reconstruction was validated with the known tracer breakthroughs from conductivity meters that were installed in the experiment volume.

The transmission data were analyzed using a novel time-lapse difference attenuation tomography routine that revealed clear areas of attenuation introduced by the tracer in the tomography plane. The results matched those from the reflection analysis well and further strengthen the validity of the approach. A first order evaluation of the resulting attenuation is in good accor-

dance with expected fracture apertures, but more research will be required for deriving more quantitative estimates of fracture apertures out of borehole GPR data. We judge our methodology to be useful to a wide range of applications well beyond flow processes in fractured crystalline rock. Examples include tracing of fluid contaminants in the subsurface and monitoring of nuclear waste repositories. Of course, it needs to be ensured that the GPR method is applicable (i.e., the electrical conductivities of the host rock need to be sufficiently low). Furthermore, it must be ensured that the measurements can be performed

quickly enough, such that the temporal behavior of the process of interest can be resolved adequately. In the field of geothermal research, we judge that applying our methodology of spatial reconstruction to GPR monitoring during stimulation experiments could strongly improve our understanding of stimulation effects in fracture networks.

*Data availability.*   The data is available under: https://www.research-collection.ethz.ch/handle/20.500.11850/456232

*Author contributions.*   P.-L. G. performed the conceptualization, investigation, formal analysis and wrote the original draft. J. D. supported

the conceptualization, investigation, and formal analysis, supervised the study, acquired funding and performed writing – review and editing. A. S. supported the formal analysis and performed writing – review and editing. M. J. supported the investigation and performed writing – review and editing. B. B. supported the investigation and performed writing – review and editing. H. M. supported the formal analysis, supervised the study and performed writing – review and editing.

*Competing interests.*   The authors declare that they have no conflict of interest.

*Acknowledgements.*   The ISC is a project of the Deep Underground Laboratory at ETH Zurich, established by the Swiss Competence Center for Energy Research - Supply of Electricity (SCCER-SoE) with the support of the Swiss Commission for Technology and Innovation (CTI). Funding for the ISC project was provided by the ETH Foundation with grants from Shell, EWZ, and by the Swiss Federal Office of Energy

through a P&D grant. Peter-Lasse Giertzuch is supported by SNF grant 200021 169894. The Grimsel Test Site is operated by Nagra, the National Cooperative for the Disposal of Radioactive Waste. We are indebted to Nagra for hosting the ISC experiment in their GTS facility and to the Nagra technical staff for onsite support.





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
