# Peer review of "4D Tracer Flow Reconstruction in Fractured Rock through Borehole GPR Monitoring"

_Solid Earth, 2020_

## Referee Comment (RC2)

The authors present a novel approach to visualizing flow in fractured rock using the combined analysis of borehlole GPR reflection profiles and cross borehole tomography. They show that even with the limited access provided by two boreholes, they are able to observe the migration of saline tracers using time-lapse radar observations. Their GPR results are supported by additional studies conducted at the same field location. The manuscript is written well and the illustrations are of good quality. My comments are primarily editorial in nature but I also have one technical concern expressed at the end of my comments. I believe the manuscript is suitable for publication after addressing the comments below.

Introduction ~line 30. On the topic of assessing the fracture surface area contributing to heat exchange, you may consider looking up the publication by Hawkins A.J., Becker M.W. and G.P. Tsoflias (2017) Evaluation of inert tracers in a bedrock fracture using ground penetrating radar and thermal sensors, Geothermics, 67, p. 86-94, http://dx.doi.org/10.1016/j.geothermics.2017.01.006.

Line 46: change "propagation of water" to "flow of water" or "infiltration of water".

Line 119: Is the INJ2.4 interval located in the plane of the GPR sections (i.e. the plane defined by boreholes Geo3 and Geo1) or is it off the plane and by how much distance? Please clarify.

Line ~135 & 140 GPR acquisition experiments 1 & 2: In the description of data acquisition, report the length of the two GPR reflection profiles, and the length of the tomographic section.

Lines 145-155: I found this part of data acquisition description difficult to comprehend and visualize. I had to read it multiple times. Maybe it is just me, but you may want to clarify better.

Line 171: Spherical spreading amplitude compensation is distance (or time) to the second power, not a linear correction.

Figure 5: Please Mark the location of the injection interval.

Figures 5 & 6 of the GPR profiles are oriented at 90 deg. rotation compared to the survey schematic depicted in figures 3 & 4, and the tomography results figures 7 & 8. I suggest figures 5 & 6 are rotated to the same orientation as the other figures so they will be easier to compare, especially the figures showing time steps fig 6 vs. fig 8.

Lines 425-429: Conductivies are reported in mS/m whereas in section 2.3 (lines 119-121) conductivities are reported in mS/cm. Please use consistent units throughout the manuscript.

Lines 430-435: I am extremely skeptical of the aperture estimates. I really do not think that the observations presented can support such detail. Tomography cannot give fraction of mm imaging. There are too many uncertainties and unknowns. Even full waveform inversion would be a stretch to provide this level of precision. Another simple question is what frequency do you use in equation 7? If you use 250 MHz (the dominant frequency of the antennas) you are overestimating frequencies. Typical GPR data is lower than the antenna dominant frequency due to attenuation. So the conductivity estimates are likely off. You'll need to provide a lot more evidence to convince me of the aperture estimates.

I think the authors have done excellent work to this point. However, estimating fracture apertures from this data is not convincing, even if the calculations give realistic results. I suggest this section is not included in the manuscript.

Thank you for your contribution. George Tsoflias

---

## Author Comment (AC1)

Dear Reviewer,

First of all, we want to thank you for your time that you have put into our contribution. We are happy to hear that you found our work interesting. Your detailed revision and constructive criticism will improve the quality of our manuscript.

Direct answers to your individual comments are found below in blue text color.

*In the manuscript, the authors describe a methodology to monitor fluid movement caused by a tracer test in granite geothermal reservoir. They apply a combination of reflection imaging and crosshole attenuation tomography to derive information on the temporal and spatial evolution of a flow field induced by a pumping experiment. Some paragraphs require language editing and should be rephrased by a native speaker. Nevertheless, the manuscript present a novel application that is of general interest to the audience and fit into the focus of this journal. Therefore, I recommend publishing this manuscript after answering to the following moderate revisions:*

We will try to improve the language quality of the manuscript in the revised version.

*Page 1, line 15ff: "Our methodology proved to be successful for characterizing flow paths related with geothermal reservoirs in crystalline rocks, but it can be transferred in a straightforward manner to other applications, such as radioactive repository monitoring or civil engineering projects."*

*I think the authors did not proved, but moreover demonstrated the applicability of the method. Furthermore, the manuscript describes not the characterization of flow path, but of tracer flow (or fluid movement), please be more specific through the manuscript.*

We will address this in the revised version and phrase this differently. Yet, we also want to emphasize here that the flow path geometry could be delineated with the method described. While classical (e.g., dye) tracer tests can characterize flow and transport, they are mostly unable to delineate the flow path geometry. We intended to highlight this advantageous feature of GPR. The new sentence will read as:

"Our methodology was demonstrated to be applicable for monitoring tracer flow and transport and characterizing flow paths related with geothermal reservoirs in crystalline rocks, but it can be transferred in a straightforward manner to other applications, such as radioactive repository monitoring or civil engineering projects."

*I believe the reader requires more background regarding the development of time-lapse*

*GPR imaging, which is yet not well covered in the introduction. Here citing a Brewster and Annan (1994) and and a conference contribution by Allroggen et al., does not cover the state of the art research in time lapse GPR imaging. I suggest to including some of the references listed in the following more recent publications:*

*Mangel, A. R., Moysey, S. M. J., & Bradford, J. (2020). Reflection tomography of timelapse GPR data for studying dynamic unsaturated flow phenomena. Hydrology and Earth System Sciences, 24(1), 159–167. https://doi.org/10.5194/hess-24-159-2020*

*Allroggen, N., Beiter, D., & Tronicke, J. (2020). Ground-penetrating radar monitoring of fast subsurface processes. Geophysics, 85(3), 1–19. https://doi.org/10.1190/geo2019-0737.1*

*Haarder, E. B., Binley, A., Looms, M. C., Doetsch, J., Nielsen, L., & Jensen, K. H. (2012).*

*Comparing Plume Characteristics Inferred from Cross-Borehole Geophysical Data. Vadose Zone Journal, 11(4), 1539–1663. https://doi.org/10.2136/vzj2012.0031*

*Allroggen, N., Garambois, S., Sénéchal, G., Rousset, D., & Tronicke, J. (2020). Crosshole reflection imaging with ground-penetrating radar data: Applications in near-surface sedimentary settings. GEOPHYSICS, 85(4), H61–H69. https://doi.org/10.1190/geo2019-0558.1*

Thank you for your comment and suggested publications. We will address this in the revised version by adding the following sentences in line 46:

"Also, the infiltration of water in unsaturated soil was successfully monitored with GPR (e.g., Trinks et al., 2001; Klenk et al., 2015). One key prerequisite of time-lapse GPR surveys is a high reproducibility and thus data consistency between the individual time steps. To this end, automated acquisition setups have been employed, such as by Mangel et al. (2020), who successfully demonstrated time-lapse reflection tomography to be capable of resolving water infiltration in the vadose zone. To resolve changes in time-lapse GPR images with higher robustness towards perturbations in the GPR traces that were unrelated to the monitored hydrological process, image similarity attributes were successfully applied to resolve fluid propagation (Allroggen et al., 2016; Allroggen et al., 2020)."

*Page 6, Line 120: "The formation water showed a conductivity of around 80 µS/cm".*

*Do you have information on the density difference of the formation water and the infiltration water. Does it make a differences for the flow formation or can the density differences be neglected?*

There is a difference in density. Formation water density was approximately 1000 g/L, while the saline tracer had an approximate density of 1030 g/L. How much difference it makes for the flow formation is uncertain, however we could not compensate for the density difference with ethanol (as described in Shakas et al. 2017) in these experiments, due to concerns about bacteria growth. We will mention this uncertainty in the revised version and make clear that comparisons with more conservative tracers need to be made with caution by adding the following in line 130:

"The salt-water-ethanol tracer that was used by Shakas et al. (2017) and Giertzuch et al. (2020) could compensate the increased density of the saline tracer in comparison with the formation water. However, in the experiments presented here, this mixture could not be used due to concerns about bacteria growth related to the ethanol. The results in Giertzuch et al. (2020) and the reflection results presented in this manuscript, do not diverge strongly, such that the effect due to the density difference is assumed to be little. However, comparisons with more conservative tracers need to be made with caution."

*Page 6, line 131: "In total, we acquired three GPR data sets…"*

*Please make sure what you mean by data set and profile. Maybe add an overview table showing the recording times and the duration of each survey?*

We will make this clearer in the revised version by adding an overview table.

*Page 7, line 170: " ...(removal of eigenvectors associated with the largest eigenvalue)."*

*How much of the data variability was removed in this process? How many eigenvectors did you remove?*

We removed only the eigenvector related to the largest eigenvalue, which mainly relates to the direct wave. The line should read "…(removal of the eigenvector associated with the largest eigenvalue)."

*Page 7, line 173: "...that was confirmed by the tomography results, other GPR surveys at the test site.."*

Something is missing in this sentence?

Yes. Thank you for noticing. The line should read "...that was confirmed by the tomography results, and other GPR surveys at the test site..."

*Page 9, line 198: "Despite the extensive correction procedures, the difference profiles still exhibited minor artifacts, resulting from improper canceling of static reflections and diffraction."*

*Similar observation have been analysed using time-lapse attributes by Allroggen et al 2016. I am not saying that you have to use such attributes, but you should at least cite this publication. Especially when presenting the SVD based filter approach.*

*Allroggen, N., & Tronicke, J. (2016). Attribute-based analysis of time-lapse ground penetrating radar data. Geophysics, 81(1), H1–H8. https://doi.org/10.1190/geo2015-0171.1*

The general problem of improper cancellation is known for difference imaging and different approaches to respond to this have been considered. Most of the signal was properly cancelled from our previously applied processing routine. This additional filter enhanced image clarity by suppressing artifacts, but this was not a key step to overcome the problems of data compatibility issues in general. We are aware of the Allroggen and Tronicke publication, which successfully imaged soil irrigation this way. However, there is no SVD based filter mentioned in this publication. Therefore, we do not judge a comparison to their approach is necessary here. Nevertheless, we will mention their approach in the revised introduction, as stated above.

*Page 9, line 202: "As for the baseline reflection processing, a time-domain Kirchhoff migration was then applied to the difference section."*

*Migration is an backpropagation of the wavefield. I do not understand how this backpropagation can be applied on the differences between two wavefields. Please add sime theoretical background (or references). To my understandung the migation should be applied before subtracting the wavefields from each other, to not introduce additional artifacts (e.g., diffraction hyperbolas )?*

We have applied the migration after the wavefield difference calculation for multiple reasons. The data differences should to our understanding be calculated on data with as little processing as possible in order to not introduce additional processing artifacts. Diffraction hyperbolas are to our understanding not an artifact, but actual data, and should thus be treated as such to calculate data differences. Some processing is necessary to retrieve compatible data sets, however migration does not help with this regard. The general application of a migration on difference data has been justified in Dorn et al. (2011), due to the linearity of the Kirchhoff migration. It has been used successfully on difference data in several studies with borehole antennas in fractured rock, such as: Dorn et al. (2011), Dorn et al. (2012a), Dorn et al. (2012b), Shakas et al. (2017), Giertzuch et al. (2020). In the revised version, we will add to line 202:

"This is possible on difference data, due to the linearity of the migration routine and makes the resulting profiles comparable to migrated GPR sections (Dorn et al., 2011). Furthermore, it helps to reduce ambient noise in the difference data due to focusing of the energy."

*Page 9, line 204: "we did not encounter significant sampling rate variations or drifts."*

*How did you the sampling rate shifts? Please add more details or remove this part from the manuscript. Furthermore, single sentences paragraphs should be merged.*

Strong sampling rate drifts can typically be observed in the raw data by trace comparisons. Additionally, Giertzuch et al. (2020) have described a method to quantify and correct for such drifts. In our experiments, such variations were not significant, and the procedure of Giertzuch et al. (2020) consequently could not improve the results. However, as the data processing routine in this manuscript closely follows that of Giertzuch et al. (2020), we judge it appropriate to mention that here we have diverged from this routine and that the reason for strong drifts is likely the control unit, rather than the antennas.

*Page 17, line 343: "Therefore, we combined the results from the two reflection surveys to at least partially overcome the radial ambiguity and confine the tracer localization:"*

*How do you partially overcome an ambiguity? Please rephrase.*

We will rephrase this in the revised version as:

"Therefore, we combined the results from the two reflection surveys to reduce the radial ambiguity and confine the tracer localization."

*Page 21, equation 6 and 7:*

*In think, you can remove the µ from the equation as it is typically close to 1 for natural materials and therefore often ignored (low loss assumption)*

We agree that µ can be removed, but with regard to the comments of the second reviewer, we have decided to remove the back-of-the-envelope calculation on fracture apertures. Hence, this part will not be included in the revised version.

*Page 21, line 421: "However, the tomography resolution and the necessary regularization makes it impossible to visualize small fractures in the results"*

*But the aim is to image fluid pathways, why to mention fractures? To my experience changes in the conductivity are images very differently than small constant features. Please rephrase.*

Virtually all fluid flow in fractured crystalline rock occurs within the fracture network, hence fluid pathways are constrained and defined through the fractures. However, with regard to the comments of the second reviewer, this part will not be included in the revised version.

*Page 21, line 435 : "…but the apertures obtained are realistic. This is an indication that our attenuation tomograms are also realistic."*

*Can you provide a reference for a realistic fracture width? What does a realistic fracture width at a single position has to do with the spatial distribution shown in the tomograms?*

The fracture aperture for the targeted fracture in the injection interval is on the order of 10^-4 m, according to Brixel et al. (2020). However, with regard to the comments of the second reviewer, this part will not be included in the revised version.

*Page 2 line 35: "...waves in MHz to GHz frequency ranges." Should read range and not ranges*

We will address this in the revised version as:

"GPR makes use of electromagnetic waves in the MHz to GHz frequency range."

*Page 2 line 26ff: usually the permittivity uses \varepsilon as a symbol*

We will address this in the revised version.

*Page 6, line 149: "In total, 38 usable reflection profiles were recorded"*

*Please add the averaged recording time of a profile.*

We will address this in the revised version, also within the suggested overview table.

*Page 7, line 164: "With the subsequently applied difference processing, temporal changes between the individual measurements can be analyzed." This sentence requires rephrasing.*

We will rephrase this in the revised version as:

"Subsequently, we applied difference processing, such that temporal changes between the individual measurements can be analyzed."

Thank you for your time and review!

Mentioned publications in our answers:

Trinks, I., H. Stümpel, and D. Wachsmuth (2001), Monitoring water flow in the unsaturated zone using georadar, *First Break*, **19**(12), DOI: 10.1046/j.1365-2397.2001.00228.x.

Klenk, P., S. Jaumann, and K. Roth (2015), Quantitative high-resolution observations of soil water dynamics in a complicated architecture using time-lapse ground-penetrating radar, *Hydrology and Earth System Sciences*, **19**(3), 1125–1139, DOI: 10.5194/hess-19-1125-2015.

Shakas, A., N. Linde, L. Baron, J. Selker, M.-F. Gerard, N. Lavenant, O. Bour, and T. Le Borgne (2017), Neutrally buoyant tracers in hydrogeophysics: Field demonstration in fractured rock, *Geophysical Research Letters*, **44**(8), 3663–3671, DOI: 10.1002/2017GL073368.

Dorn, C., N. Linde, T. Le Borgne, O. Bour, and L. Baron (2011), Single-hole GPR reflection imaging of solute transport in a granitic aquifer, *Geophysical Research Letters*, **38**(8), DOI: 10.1029/2011GL047152.

Dorn, C., N. Linde, T. Le Borgne, O. Bour, and M. Klepikova (2012a), Inferring transport characteristics in a fractured rock aquifer by combining single-hole ground-penetrating radar reflection monitoring and tracer test data, *Water Resources Research*, **48**(11), DOI: 10.1029/2011WR011739.

Dorn, C., N. Linde, J. Doetsch, T. Le Borgne, and O. Bour (2012b), Fracture imaging within a granitic rock aquifer using multiple-offset single-hole and cross-hole GPR reflection data, *Journal of Applied Geophysics*, **78**, 123–132, DOI: 10.1016/j.jappgeo.2011.01.010.

Giertzuch, P.-L., J. Doetsch, M. Jalali, A. Shakas, C. Schmelzbach, and H. Maurer (2020), Time-lapse ground penetrating radar difference reflection imaging of saline tracer flow in fractured rock, *Geophysics*, **85**(3), H25–H37, DOI: 10.1190/geo2019-0481.1.

Brixel, B., M. Klepikova, Q. Lei, C. Roques, M. R. Jalali, H. Krietsch, and S. Loew (2020), Tracking Fluid Flow in Shallow Crustal Fault Zones: 2. Insights From Cross-Hole Forced Flow Experiments in Damage Zones, *Journal of Geophysical Research: Solid Earth*, **125**(4), e2019JB019,108, DOI: 10.1029/2019JB019108.

---

## Author Comment (AC2)

Dear George Tsoflias,

Thank you for your revision and constructive criticism on our manuscript. We are happy to hear that you found our work interesting and the manuscript well written.

We have decided with regard to your concern to exclude the part about the aperture estimation in the discussion section. We believe that you are correct and it does not strengthen our manuscript.

Direct answers to your individual comments are found below in blue text color.

*The authors present a novel approach to visualizing flow in fractured rock using the combined analysis of borehole GPR reflection profiles and cross borehole tomography. They show that even with the limited access provided by two boreholes, they are able to observe the migration of saline tracers using time-lapse radar observations. Their GPR results are supported by additional studies conducted at the same field location. The manuscript is written well and the illustrations are of good quality. My comments are primarily editorial in nature but I also have one technical concern expressed at the end of my comments. I believe the manuscript is suitable for publication after addressing the comments below.*

*Introduction ~line 30. On the topic of assessing the fracture surface area contributing to heat exchange, you may consider looking up the publication by Hawkins A.J., Becker M.W. and G.P. Tsoflias (2017) Evaluation of inert tracers in a bedrock fracture using ground penetrating radar and thermal sensors, Geothermics, 67, p. 86-94, http://dx.doi.org/10.1016/j.geothermics.2017.01.006.*

We will mention the publication in the revised version by adding in line 31:

"Hawkins et al. (2017) reported significant differences in heat transport related to flow channeling within a fracture that was monitored with time-lapse GPR."

*Line 46: change "propagation of water" to "flow of water" or "infiltration of water".*

Yes, we will address this in the revised version.

*Line 119: Is the INJ2.4 interval located in the plane of the GPR sections (i.e. the plane defined by boreholes Geo3 and Geo1) or is it off the plane and by how much distance? Please clarify.*

The INJ2.4 interval is located approximately 1.7m below the GEO1-GEO3 borehole plane. We will clarify this in the revised version by writing in line 119:

"A saline tracer with a conductivity of approximately 60 mS/cm was injected at a constant flow rate of approximately 2 L/min in the INJ2.4 interval, which is located in between the S3 shear zones, approximately 1.7 m below the GEO1-GEO3 plane (Figure 2)."

*Line ~135 & 140 GPR acquisition experiments 1 & 2: In the description of data acquisition, report the length of the two GPR reflection profiles, and the length of the tomographic section.*

The boreholes had a length of 30m, which was fully used. We will clarify this in the revised version in the lines 136 and 141, and will add a table to provide a better overview on the experiments and respective GPR surveys. The used antenna positions in the tomography are presented in Figure 4.

*Lines 145-155: I found this part of data acquisition description difficult to comprehend and visualize. I had to read it multiple times. Maybe it is just me, but you may want to clarify better.*

We will try to clarify this in the revised version.

*Line 171: Spherical spreading amplitude compensation is distance (or time) to the second power, not a linear correction.*

Contrary to a more common spherical spreading and attenuation correction with r^2, we took a combined approach of compensating for spherical spreading and attenuation. To our understanding, the energy decreases with r^2 due to spherical spreading, hence the amplitude should be decreasing linearly with r. Additionally, we then applied an exponential gain to compensate for the wave attenuation.

*Figure 5: Please Mark the location of the injection interval.*

We will do this in the revised version.

*Figures 5 & 6 of the GPR profiles are oriented at 90 deg. rotation compared to the survey schematic depicted in figures 3 & 4, and the tomography results figures 7 & 8. I suggest figures 5 & 6 are rotated to the same orientation as the other figures so they will be easier to compare, especially the figures showing time steps fig 6 vs. fig 8.*

We will do this in the revised version.

*Lines 425-429: Conductivies are reported in mS/m whereas in section 2.3 (lines 119-121) conductivities are reported in mS/cm. Please use consistent units throughout the manuscript.*

Thank you for noticing. With regard to your comment about the apertures estimation, this part will not be included in the revised version.

*Lines 430-435: I am extremely skeptical of the aperture estimates. I really do not think that the observations presented can support such detail. Tomography cannot give fraction of mm imaging. There are too many uncertainties and unknowns. Even full waveform inversion would be a stretch to provide this level of precision. Another simple question is what frequency do you use in equation 7? If you use 250 MHz (the dominant frequency of the antennas) you are overestimating frequencies. Typical GPR data is lower than the antenna dominant frequency due to attenuation. So the conductivity estimates are likely off. You'll need to provide a lot more evidence to convince me of the aperture estimates.*

*I think the authors have done excellent work to this point. However, estimating fracture apertures from this data is not convincing, even if the calculations give realistic results. I suggest this section is not included in the manuscript.*

With regard to your concerns, we have decided to exclude this approximation from our manuscript.

*Thank you for your contribution. George Tsoflias*

Thank you for your time and review!

---

## Author Response (AR1)

Dear Editor and Reviewers,

Thank you for your valuable feedback and comments on our manuscript. All comments were helpful and could improve the quality of our manuscript. Detailed point-by-point answers are found below in blue text color. The line numbers refer to the old version of the manuscript. During the last proofreading some of the planned text edits that were expressed in the individual author comments have been slightly adapted for the revised manuscript. All changes are indicated in the track-changes file.

Peter-Lasse Giertzuch
on behalf of all authors

**Reviewer 1**

*In the manuscript, the authors describe a methodology to monitor fluid movement caused by a tracer test in granite geothermal reservoir. They apply a combination of reflection imaging and crosshole attenuation tomography to derive information on the temporal and spatial evolution of a flow field induced by a pumping experiment. Some paragraphs require language editing and should be rephrased by a native speaker. Nevertheless, the manuscript present a novel application that is of general interest to the audience and fit into the focus of this journal. Therefore, I recommend publishing this manuscript after answering to the following moderate revisions:*

We have tried to improve the language quality of the manuscript by careful proofreading.

*Page 1, line 15ff: "Our methodology proved to be successful for characterizing flow paths related with geothermal reservoirs in crystalline rocks, but it can be transferred in a straightforward manner to other applications, such as radioactive repository monitoring or civil engineering projects."*

*I think the authors did not proved, but moreover demonstrated the applicability of the method. Furthermore, the manuscript describes not the characterization of flow path, but of tracer flow (or fluid movement), please be more specific through the manuscript.*

We have addressed this in the revised version and phrased this differently. Yet, we want to emphasize here that also the flow path geometry could be delineated with the method described. While classical (e.g., dye) tracer tests can characterize flow and transport, they are mostly unable to delineate the flow path geometry. We intended to highlight this advantageous feature of GPR. The new sentence reads as:

"Our methodology was demonstrated to be applicable for monitoring tracer flow and transport and characterizing flow paths related with geothermal reservoirs in crystalline rocks, but it can be transferred in a straightforward manner to other applications, such as radioactive repository monitoring or civil engineering projects."

*I believe the reader requires more background regarding the development of time-lapse*

*GPR imaging, which is yet not well covered in the introduction. Here citing a Brewster and Annan (1994) and and a conference contribution by Allroggen et al., does not cover the state of the art research in time lapse GPR imaging. I suggest to including some of the references listed in the following more recent publications:*

*Mangel, A. R., Moysey, S. M. J., & Bradford, J. (2020). Reflection tomography of timelapse GPR data for studying dynamic unsaturated flow phenomena. Hydrology and Earth System Sciences, 24(1), 159–167. https://doi.org/10.5194/hess-24-159-2020*

*Allroggen, N., Beiter, D., & Tronicke, J. (2020). Ground-penetrating radar monitoring of fast subsurface processes. Geophysics, 85(3), 1–19. https://doi.org/10.1190/geo2019-0737.1*

*Haarder, E. B., Binley, A., Looms, M. C., Doetsch, J., Nielsen, L., & Jensen, K. H. (2012).*

*Comparing Plume Characteristics Inferred from Cross-Borehole Geophysical Data. Vadose Zone Journal, 11(4), 1539–1663. https://doi.org/10.2136/vzj2012.0031*

*Allroggen, N., Garambois, S., Sénéchal, G., Rousset, D., & Tronicke, J. (2020). Crosshole reflection imaging with ground-penetrating radar data: Applications in near-surface sedimentary settings. GEOPHYSICS, 85(4), H61–H69. https://doi.org/10.1190/geo2019-0558.1*

Thank you for your comment and suggested publications. We have addressed this in the revised version by adding the following sentences in line 46:

"Also, the infiltration of water in unsaturated soil was successfully monitored with GPR (e.g., Trinks et al., 2001; Klenk et al., 2015). One key prerequisite of time-lapse GPR surveys is a high reproducibility and thus data consistency between the individual time steps. To this end, automated acquisition setups have been employed, such as that used by Mangel et al. (2020), who successfully demonstrated time-lapse reflection tomography to be capable of resolving water infiltration in the vadose zone. To resolve changes in time-lapse GPR images with higher robustness towards perturbations in the GPR traces that were unrelated to the monitored hydrological process, image similarity attributes were successfully applied (Allroggen and Tronicke, 2016; Allroggen et al., 2020)."

*Page 6, Line 120: "The formation water showed a conductivity of around 80 μS/cm".*

*Do you have information on the density difference of the formation water and the infiltration water. Does it make a differences for the flow formation or can the density differences be neglected?*

There is a difference in density. Formation water density was approximately 1000 g/L, while the saline tracer had an approximate density of 1030 g/L. How much difference it makes for the flow formation is uncertain, but we could not compensate for the density difference with ethanol (as described in Shakas et al. 2017) in our experiments, due to concerns about bacteria growth. We have mentioned this uncertainty in the revised version and made clear that comparisons with more conservative tracers need to be made with caution by adding the following in line 130:

"The salt-water-ethanol tracer that was used by Shakas et al. (2017) and Giertzuch et al. (2020) could compensate the increased density of the saline tracer in comparison to the formation water, but in the experiments presented here, this mixture could not be used due to concerns about bacteria growth related to the ethanol. Since the results presented in Giertzuch et al. (2020) showed comparable tracer appearances as the reflection results in this manuscript, the effect due to the density difference is assumed to be small. However, for comparisons with more conservative tracers the density difference should be noted."

*Page 6, line 131: "In total, we acquired three GPR data sets…"*

*Please make sure what you mean by data set and profile. Maybe add an overview table showing the recording times and the duration of each survey?*

We have made this clearer in the revised version by adding the suggested overview table. Also we have rephrased line 131: "In total, we performed three GPR surveys, two of them during the tracer experiments and one transmission GPR survey in the unperturbed experiment volume."

*Page 7, line 170: " ...(removal of eigenvectors associated with the largest eigenvalue)."*

*How much of the data variability was removed in this process? How many eigenvectors did you remove?*

We removed only the eigenvector related to the largest eigenvalue, which mainly relates to the direct wave. The line now reads "…(removal of the eigenvector associated with the largest eigenvalue)."

*Page 7, line 173: "...that was confirmed by the tomography results, other GPR surveys at the test site.."*

Something is missing in this sentence?

Yes. Thank you for noticing. The line now reads "...that was confirmed by the tomography results, and other GPR surveys at the test site..."

*Page 9, line 198: "Despite the extensive correction procedures, the difference profiles still exhibited minor artifacts, resulting from improper canceling of static reflections and diffraction."*

*Similar observation have been analysed using time-lapse attributes by Allroggen et al 2016. I am not saying that you have to use such attributes, but you should at least cite this publication. Especially when presenting the SVD based filter approach.*

*Allroggen, N., & Tronicke, J. (2016). Attribute-based analysis of time-lapse ground penetrating radar data. Geophysics, 81(1), H1–H8. https://doi.org/10.1190/geo2015-0171.1*

The general problem of improper cancellation is known for difference imaging and different approaches to address this have been considered. Most of the signal was properly cancelled from our previously applied processing routine. This additional filter enhanced image clarity by suppressing artifacts, but this was not a key step to overcome the problems of data compatibility issues in general. We are aware of the Allroggen and Tronicke publication, which successfully imaged soil irrigation this way. However, there is no SVD based filter mentioned in this publication. Therefore, we do not judge a comparison to their approach is necessary here. Nevertheless, we have mentioned their approach in the revised introduction, as stated above.

*Page 9, line 202: "As for the baseline reflection processing, a time-domain Kirchhoff migration was then applied to the difference section."*

*Migration is an backpropagation of the wavefield. I do not understand how this backpropagation can be applied on the differences between two wavefields. Please add sime theoretical background (or references). To my understandung the migation should be applied before subtracting the wavefields from each other, to not introduce additional artifacts (e.g., diffraction hyperbolas )?*

We have applied the migration after the wavefield difference calculation for multiple reasons. The data differences should to our understanding be calculated on data with as little processing as possible in order to not introduce additional processing artifacts. Diffraction hyperbolas are to our understanding

not an artifact, but actual data, and should thus be treated as such to calculate data differences. Some processing is necessary to retrieve compatible data sets, but migration does not help with this regard. The general application of a migration on difference data has been justified in Dorn et al. (2011), due to the linearity of the Kirchhoff migration. It has been used successfully on difference data in several studies with borehole antennas in fractured rock, such as: Dorn et al. (2011), Dorn et al. (2012a), Dorn et al. (2012b), Shakas et al. (2017), Giertzuch et al. (2020). In the revised version, we will add to line 202:

"This is possible on difference data, due to the linearity of the migration routine, and makes the resulting profiles comparable to migrated GPR sections (Dorn et al., 2012). Furthermore, it helps to reduce ambient noise in the difference data due to focusing of the energy."

*Page 9, line 204: "we did not encounter significant sampling rate variations or drifts."*

*How did you the sampling rate shifts? Please add more details or remove this part from the manuscript. Furthermore, single sentences paragraphs should be merged.*

Strong sampling rate drifts can typically be observed in the raw data by trace comparisons. Additionally, Giertzuch et al. (2020) have described a method to quantify and correct for such drifts. In our experiments, such variations were not significant, and the procedure of Giertzuch et al. (2020) consequently could not improve the results. However, as the data processing routine in this manuscript closely follows that of Giertzuch et al. (2020), we judge it appropriate to mention that here we have diverged from this routine and that the reason for strong drifts is likely the control unit, rather than the antennas. We have added the following sentence in line 205:

"This was determined with the procedure described by Giertzuch et al. (2020) and was unexpected, because the same antennas were used in this survey."

*Page 17, line 343: "Therefore, we combined the results from the two reflection surveys to at least partially overcome the radial ambiguity and confine the tracer localization:"*

*How do you partially overcome an ambiguity? Please rephrase.*

We have rephrased this in the revised version as:

 "Therefore, we combined the results from the two reflection surveys to reduce the radial ambiguity and confine the tracer localization:"

*Page 21, equation 6 and 7:*

*In think, you can remove the μ from the equation as it is typically close to 1 for natural materials and therefore often ignored (low loss assumption)*

We agree that μ can be removed, but with regard to the comments of the second reviewer, we have decided to remove the back-of-the-envelope calculation on fracture apertures. Hence, this part is not included in the revised version.

*Page 21, line 421: "However, the tomography resolution and the necessary regularization makes it impossible to visualize small fractures in the results"*

*But the aim is to image fluid pathways, why to mention fractures? To my experience changes in the conductivity are images very differently than small constant features. Please rephrase.*

Virtually all fluid flow in fractured crystalline rock occurs within the fracture network, hence fluid pathways are constrained and defined through the fractures. However, with regard to the comments of the second reviewer, this part is not included in the revised version.

*Page 21, line 435 : "…but the apertures obtained are realistic. This is an indication that our attenuation tomograms are also realistic."*

*Can you provide a reference for a realistic fracture width? What does a realistic fracture width at a single position has to do with the spatial distribution shown in the tomograms?*

The fracture aperture for the targeted fracture in the injection interval is on the order of $10^{-4}$ m, according to Brixel et al. (2020). However, with regard to the comments of the second reviewer, this part is not included in the revised version.

*Page 2 line 35: "…waves in MHz to GHz frequency ranges." Should read range and not ranges*

We have addressed this in the revised version as:

"GPR makes use of electromagnetic waves in the MHz to GHz frequency range."

*Page 2 line 26ff: usually the permittivity uses \varepsilon as a symbol*

We have addressed this in the revised version.

*Page 6, line 149: "In total, 38 usable reflection profiles were recorded"*

*Please add the averaged recording time of a profile.*

We have addressed this in the revised version in the respective line and also within the suggested overview table.

*Page 7, line 164: "With the subsequently applied difference processing, temporal changes between the individual measurements can be analyzed." This sentence requires rephrasing.*

We have rephrased this in the revised version as:

"Subsequently, we applied difference processing, such that temporal changes between the individual measurements can be analyzed."

Thank you for your time and review!

Mentioned publications in our answers:

Trinks, I., H. Stümpel, and D. Wachsmuth (2001), Monitoring water flow in the unsaturated zone using georadar, *First Break*, **19**(12), DOI: 10.1046/j.1365-2397.2001.00228.x.

Klenk, P., S. Jaumann, and K. Roth (2015), Quantitative high-resolution observations of soil water dynamics in a complicated architecture using time-lapse ground-penetrating radar, *Hydrology and Earth System Sciences*, **19**(3), 1125–1139, DOI: 10.5194/hess-19-1125-2015.

Shakas, A., N. Linde, L. Baron, J. Selker, M.-F. Gerard, N. Lavenant, O. Bour, and T. Le Borgne (2017),

Neutrally buoyant tracers in hydrogeophysics: Field demonstration in fractured rock, *Geophysical Research Letters*, **44**(8), 3663–3671, DOI: 10.1002/2017GL073368.

Dorn, C., N. Linde, T. Le Borgne, O. Bour, and L. Baron (2011), Single-hole GPR reflection imaging of solute transport in a granitic aquifer, *Geophysical Research Letters*, **38**(8), DOI: 10.1029/2011GL047152.

Dorn, C., N. Linde, T. Le Borgne, O. Bour, and M. Klepikova (2012a), Inferring transport characteristics in a fractured rock aquifer by combining single-hole ground-penetrating radar reflection monitoring and tracer test data, *Water Resources Research*, **48**(11), DOI: 10.1029/2011WR011739.

Dorn, C., N. Linde, J. Doetsch, T. Le Borgne, and O. Bour (2012b), Fracture imaging within a granitic rock aquifer using multiple-offset single-hole and cross-hole GPR reflection data, *Journal of Applied Geophysics*, **78**, 123–132, DOI: 10.1016/j.jappgeo.2011.01.010.

Giertzuch, P.-L., J. Doetsch, M. Jalali, A. Shakas, C. Schmelzbach, and H. Maurer (2020), Time-lapse ground penetrating radar difference reflection imaging of saline tracer flow in fractured rock, *Geophysics*, **85**(3), H25–H37, DOI: 10.1190/geo2019-0481.1.

Brixel, B., M. Klepikova, Q. Lei, C. Roques, M. R. Jalali, H. Krietsch, and S. Loew (2020), Tracking Fluid Flow in Shallow Crustal Fault Zones: 2. Insights From Cross-Hole Forced Flow Experiments in Damage Zones, *Journal of Geophysical Research: Solid Earth*, **125**(4), e2019JB019,108, DOI: 10.1029/2019JB019108.

**Reviewer 2:**

*The authors present a novel approach to visualizing flow in fractured rock using the combined analysis of borehlole GPR reflection profiles and cross borehole tomography. They show that even with the limited access provided by two boreholes, they are able to observe the migration of saline tracers using time-lapse radar observations. Their GPR results are supported by additional studies conducted at the same field location. The manuscript is written well and the illustrations are of good quality. My comments are primarily editorial in nature but I also have one technical concern expressed at the end of my comments. I believe the manuscript is suitable for publication after addressing the comments below.*

*Introduction ~line 30. On the topic of assessing the fracture surface area contributing to heat exchange, you may consider looking up the publication by Hawkins A.J., Becker M.W. and G.P. Tsoflias (2017) Evaluation of inert tracers in a bedrock fracture using ground penetrating radar and thermal sensors, Geothermics, 67, p. 86-94, http://dx.doi.org/10.1016/j.geothermics.2017.01.006.*

We have mentioned the publication in the revised version by adding in line 31:

"Hawkins et al. (2017) reported significant differences in heat transport related to flow channeling within a fracture that was monitored with time-lapse GPR."

*Line 46: change "propagation of water" to "flow of water" or "infiltration of water".*

Yes, we have addressed this in the revised version as "infiltration of water".

*Line 119: Is the INJ2.4 interval located in the plane of the GPR sections (i.e. the plane defined by boreholes Geo3 and Geo1) or is it off the plane and by how much distance? Please clarify.*

The INJ2.4 interval is located approximately 1.7m below the GEO1-GEO3 borehole plane. We have clarified this in the revised version by writing in line 119:

"A saline tracer with a conductivity of approximately 60 mS/cm was injected at a constant flow rate of approximately 2 L/min in the INJ2.4 interval, which is located in between the S3 shear zones, approximately 1.7 m below the GEO1-GEO3 plane (Figure 2)."

*Line ~135 & 140 GPR acquisition experiments 1 & 2: In the description of data acquisition, report the length of the two GPR reflection profiles, and the length of the tomographic section.*

The boreholes had a length of 30m, which was fully used. We have clarified this in the revised version in the lines 136 and 141, and added a table to provide a better overview on the experiments and respective GPR surveys. The used antenna positions in the tomography are presented in Figure 4.

*Lines 145-155: I found this part of data acquisition description difficult to comprehend and visualize. I had to read it multiple times. Maybe it is just me, but you may want to clarify better.*

We have tried to clarify this in the revised version by rewriting line 145ff:

"Simultaneously during experiment 2, a dual-channel transmission survey was conducted, between the static antennas in GEO3 and the moving antennas in GEO1. The two transmission channels (see Figure 3c) were triggered every 20 cm, while moving the antenna array in GEO1 upwards, with the static antenna array in GEO3 at a fixed position. After each of these recordings, the antenna array in GEO3 was moved to a new position, to record another multi-offset gather. In total, eight different static antenna array positions were occupied in GEO3. To reduce the data acquisition time for the tomography sections, we exploited the potential of the dual-channel setup with two antenna arrays by alternating between two acquisition sets. The positions of the single antennas in the static array, which were separated by 5 m, were chosen to be placed as presented in Table 2. Set 1 covered the positions between 0m and 17.5 m, and Set 2 covered the positions between 5m and 22.5m reference from the bottom of the borehole. The straight ray patterns of the two deployments are shown in Figures 4b and 4c. With this procedure, it was possible to use the data either as a more comprehensive data set with all 16 positions but a lower time resolution, or with only eight positions but a higher time resolution. Both of these options were later used to invert for the change in attenuation.

The acquisition of each of the (full) transmission data sets took about 40 min. During that time, eight reflection profiles were recorded simultaneously. The GPR survey lasted for eight hours, but during that time the antennas had to be recharged. In total, one of the two transmission channels recorded eight full sets, the other seven during the experiment. Additionally, one full transmission data set was recorded prior to the tracer injection to serve as the reference data (Figure 3c)."

*Line 171: Spherical spreading amplitude compensation is distance (or time) to the second power, not a linear correction.*

Contrary to a more common spherical spreading and attenuation correction with $r^2$, we took an approach of compensating for spherical spreading and attenuation separately. To our understanding, the energy decreases with $r^2$ due to spherical spreading, hence the amplitude should be decreasing linearly with r. Additionally, we then applied an exponential gain to compensate for the wave attenuation.

*Figure 5: Please Mark the location of the injection interval.*

We have done this in the revised version.

*Figures 5 & 6 of the GPR profiles are oriented at 90 deg. rotation compared to the survey schematic depicted in figures 3 & 4, and the tomography results figures 7 & 8. I suggest figures 5 & 6 are rotated to the same orientation as the other figures so they will be easier to compare, especially the figures showing time steps fig 6 vs. fig 8.*

We have done this in the revised version and also updated the supplementary material accordingly.

*Lines 425-429: Conductivies are reported in mS/m whereas in section 2.3 (lines 119-121) conductivities are reported in mS/cm. Please use consistent units throughout the manuscript.*

Thank you for noticing. With regard to your comment about the apertures estimation, this part is not included in the revised version. We have removed the lines 415-437, lines 474-476, and removed Figure 12.

*Lines 430-435: I am extremely skeptical of the aperture estimates. I really do not think that the observations presented can support such detail. Tomography cannot give fraction of mm imaging. There are too many uncertainties and unknowns. Even full waveform inversion would be a stretch to provide this level of precision. Another simple question is what frequency do you use in equation 7? If you use 250 MHz (the dominant frequency of the antennas) you are overestimating frequencies. Typical GPR data is lower than the antenna dominant frequency due to attenuation. So the conductivity estimates are likely off. You'll need to provide a lot more evidence to convince me of the aperture estimates.*

*I think the authors have done excellent work to this point. However, estimating fracture apertures from this data is not convincing, even if the calculations give realistic results. I suggest this section is not included in the manuscript.*

With regard to your concerns, we have decided to exclude this approximation from our manuscript. We have removed the lines 415-437, lines 474-476, and removed Figure 12.

*Thank you for your contribution. George Tsoflias*

Thank you for your time and review!

---

## Author Response (AR2)

Dear Editorial Team of Solid Earth, dear Reviewer,

Thank you for your remark for a technical correction on our manuscript. Our answer is found below along with the changes we have made to the text. Additionally, please note that the affiliation of the author Bernard Brixel needed to be changed into Geological Institute, ETH Zurich, Zurich, Switzerland.

On behalf of all authors,
Peter Giertzuch

*The authors have greatly improved the quality of the manuscript and responded well to the previous comments. I have only one remaining comment, before the manuscript can be accepted for publication.*

*Page 11, line 228:*

*"This is possible on difference data, due to the linearity of the migration routine, and makes the resulting profiles comparable to migrated GPR sections (Dorn et al., 2012)."*

*Please explain briefly what you mean by "linearity of the migration routine" and the underlying assumptions. I believe you assume constant GPR velocity between both reference GPR wavefields?*

The underlying assumption to apply the Kirchhoff migration on differenced data, is that the migration operator is a linear operator. This is what we meant with "linearity of the migration routine". The GPR velocity is also assumed to stay constant, however this is not the reasoning behind the approach. We tried to make this clearer now, by rephrasing on Page 11, line 228:

*Migrating difference data is useful, since Kirchhoff migration is a linear operator and makes the resulting profiles comparable to migrated GPR sections (Dorn et al., 2012).*

Additionally, we have added the information on the used migration routine on Page 9, line 196:

*Finally, the reflection sections underwent a Kirchhoff migration (from the CREWES Matlab package, Margrave and Lamoureux, 2019) using a constant velocity of 0.12 m/ns that was confirmed by the tomography results, and other GPR surveys at the test site (Giertzuch et al., 2020; Doetsch et al., 2020).*